# YAP-dependent necrosis occurs in early stages of Alzheimer's disease and regulates mouse model pathology

Hikari Tanaka[1,14], Hidenori Homma[1,14], Kyota Fujita[1,14], Kanoh Kondo[1,14], Shingo Yamada[2,14], Xiaocen Jin[1], Masaaki Waragai [3], Gaku Ohtomo[4], Atsushi Iwata[4], Kazuhiko Tagawa[1], Naoki Atsuta[5], Masahisa Katsuno [5], Naoki Tomita[6], Katsutoshi Furukawa[6], Yuko Saito[7], Takashi Saito[8], Ayaka Ichise[9], Shinsuke Shibata [9], Hiroyuki Arai[6], Takaomi Saido[8], Marius Sudol[10], Shin-ichi Muramatsu [11], Hideyuki Okano[9], Elliott J. Mufson[12], Gen Sobue [5], Shigeo Murayama[13] & Hitoshi Okazawa[1]*

The timing and characteristics of neuronal death in Alzheimer's disease (AD) remain largely unknown. Here we examine AD mouse models with an original marker, myristoylated alanine-rich C-kinase substrate phosphorylated at serine 46 (pSer46-MARCKS), and reveal an increase of neuronal necrosis during pre-symptomatic phase and a subsequent decrease during symptomatic phase. Postmortem brains of mild cognitive impairment (MCI) rather than symptomatic AD patients reveal a remarkable increase of necrosis. In vivo imaging reveals instability of endoplasmic reticulum (ER) in mouse AD models and genome-edited human AD iPS cell-derived neurons. The level of nuclear Yes-associated protein (YAP) is remarkably decreased in such neurons under AD pathology due to the sequestration into cytoplasmic amyloid beta (Aβ) aggregates, supporting the feature of YAP-dependent necrosis. Suppression of early-stage neuronal death by AAV-YAPdeltaC reduces the later-stage extracellular Aβ burden and cognitive impairment, suggesting that preclinical/pro-dromal YAP-dependent neuronal necrosis represents a target for AD therapeutics.

[1] Department of Neuropathology, Medical Research Institute and Center for Brain Integration Research, Tokyo Medical and Dental University, 1-5-45 Yushima, Bunkyo-ku, Tokyo 113-8510, Japan. [2] Shino-Test Corporation, 2-29-14, Ohino-dai, Minami-ku, Sagamihara, Kanagawa 252-0331, Japan. [3] Department of Neurology, Higashi Matsudo Municipal Hospital, Matsudo, Chiba 270-2222, Japan. [4] Department of Neurology, The University of Tokyo, Graduate School of Medicine, 7-3-1 Hongo, Bunkyo-ku, Tokyo 113-8655, Japan. [5] Department of Neurology, Brain and Mind Research Center, Nagoya University Graduate School of Medicine, 65 Tsurumai-cho, Showa-ku, Nagoya Aichi 466-8550, Japan. [6] Department of Geriatrics & Gerontology, Division of Brain Science, Institute of Development, Aging and Cancer, Tohoku University, 4-1, Seiryo-cho, Aoba-ku, Sendai 980-8575, Japan. [7] Department of Laboratory Medicine, National Center Hospital, National Center of Neurology and Psychiatry, 4-1-1 Ogawa-Higashi-machi, Kodaira, Tokyo, Japan. [8] Laboratory for Proteolytic Neuroscience, RIKEN Center for Brain Science, 2-1 Hirosawa, Wako, Saitama 351-0198, Japan. [9] Department of Physiology, Keio University School of Medicine, 35 Shinano-machi, Shinjuku-ku, Tokyo 160-8582, Japan. [10] Department of Physiology, National University of Singapore, Yong Loo Li School of Medicine, 2 Medical Drive, Singapore 117597, Singapore. [11] Department of Neurology, Jichi Medical University, 3311-1 Yakushiji, Shimotsuke, Tochigi 329-0496, Japan. [12] Department of Neurobiology and Neurology, Barrow Neurological Institute, 350 W. Thomas Road, Phoenix, AZ 85013, USA. [13] Department of Neuropathology, Brain Bank for Aging Research, Tokyo Metropolitan Institute of Gerontology, 35-2, Sakae-cho, Itabashi-ku, Tokyo 173-0015, Japan. [14] These authors contributed equally: Hikari Tanaka, Hidenori Homma, Kyota Fujita, Kanoh Kondo, Shingo Yamada. *email: okazawa.npat@mri.tmd.ac.jp

The ability to diagnose AD at an early stage is eagerly anticipated, especially after clinical trials of anti-Aβ antibodies[1,2] and γ-/β-secretase inhibitors[3,4] in post-onset patients proved disappointing. A deeper understanding of MCI could play a pivotal role in the development of new therapeutic strategies for AD. Despite the importance of MCI, the pathological and molecular evaluation remains insufficient especially from the aspect of chronological change of neuronal function and cell death. Accordingly, no efficient single biomarker directly reflecting disease activity in MCI has yet been reported.

Cutting-edge techniques, including comprehensive analyses, have identified molecules in addition to Aβ and tau that could be targeted for therapeutic intervention at the early stage of AD. For instance, comparison of neuroimaging and transcriptome data revealed that a genetic profile of lipid metabolism centered by APOE affects propagation patterns of both Aβ and tau in the brain[5]. In another study, a meta-analysis of functional genomic data from AD showed that YAP, a co-transcriptional factor that regulates cell death and survival by binding to the different transcription factors p73 and TEA domain family member 1 (TEAD)[6–9], is positioned at the center of the molecular network of AD[10]. Elevated activity of TEAD mediated by YAP has been implicated in cell proliferation, differentiation, and survival[11–13], whereas elevated p73[14–16] activity and reduced TEAD[17–19] activity promote apoptosis and necrosis, respectively.

Previously, we performed a comprehensive phosphoproteome analysis of four strains of AD model mice and human postmortem AD brains, and discovered three proteins whose phosphorylation state is altered at a very early stage before extracellular amyloid aggregates[20]. One such protein is MARCKS, which anchors the actin cytoskeleton to the plasma membrane and plays a critical role in stabilizing the post-synaptic structure of dendritic spines[21]. Phosphorylation of MARCKS at Ser46 decreases its affinity for actin and destabilizes dendritic spines[22]. High mobility group box-1 (HMGB1) contributes to the MARCKS phosphorylation via Toll-like receptor 4 (TLR4) since blockade of HMGB1–TLR4 binding with monoclonal anti-HMGB1 antibodies suppresses the phosphorylation of MARCKS at Ser46, stabilizes dendritic spines, and rescues cognitive impairment in AD model mice[22]. Given that HMGB1 is released from necrotic cells[23,24], it remains unclear how MARCKS phosphorylation, which occurs at the early stage of AD pathology, is connected to neuronal cell death, which is believed to occur at a relatively late stage.

In this study, we found that HMGB1 levels were remarkably elevated in CSF of MCI, but not so elevated in AD patients. Consistent with this, active neuronal necrosis revealed by pSer46-MARCKS increased to the greatest extent during preclinical stages of AD mouse models and human MCI patients. In addition, we showed that the observed necrosis was caused by a deficiency of YAP, resulting in suppression of the transcriptional activity of TEAD, the final effector molecule of the Hippo pathway[11–13], in mouse AD models, human AD iPS neuron models and human postmortem MCI brains. These findings unravel the occurrence of cell death at the early stage in AD, which could be a therapeutic target that prevents progression of AD.

## Results

**HMGB1 is elevated in CSF of human MCI patients**. CSF samples were collected by lumbar puncture from 34 normal controls, 14 disease controls, 26 MCI patients, and 73 AD patients (Supplementary Tables 1, 2). MCI and AD were diagnosed by ICD-10, and the patients were categorized as having amnestic MCI. There was no significant difference in age between the different patient groups, but the proportion of female patients was slightly higher in the AD group than in the other groups (Supplementary

Table 1). ApoE subtyping was performed in 19 MCI and 18 AD patients (Supplementary Table 1). In the disease control patients, CSF samples were taken because neurological diseases were suspected; therefore, there was some bias in the types of disease present in this patient group (Supplementary Table 2). To verify the accuracy of MCI/AD diagnosis, we compared the levels of Aβ42, pTau, and pTau/Aβ42 between the normal control group and the MCI or AD group. In support of the clinical diagnoses, Aβ42 levels were reduced, and pTau/Aβ42 levels were elevated, in the MCI and AD groups (Supplementary Fig. 1). APP/Aβ ratio was increased in AD group in comparison to other groups (Supplementary Fig. 1), as reported previously[25].

Expecting elevation of HMGB1 in symptomatic AD, we evaluated HMGB1 concentrations in CSF by ELISA. However, the CSF-HMGB1 level was significantly elevated in the clinically diagnosed MCI group, but not the AD group, relative to the normal or disease controls (Fig. 1a). The CSF-HMGB1 level was also significantly higher in the MCI group than in the AD group (Fig. 1a). In receiver operating characteristic (ROC) analysis of the comparisons between MCI and the normal or disease controls, the area under the curve (AUC) was 0.861 or 0.931, respectively (Fig. 1b). In addition, AUC was 0.809 in comparison between MCI and AD, suggesting the CSF-HMGB1 value may assist clinical diagnosis of the two phenotypic states (Fig. 1b). Interestingly, we observed no significant correlations between CSF-HMGB1 and Mini-Mental State Examination (MMSE) score in the MCI, AD, or MCI + AD group (Supplementary Fig. 2).

In the MCI group, we observed a positive relationship between levels of CSF-HMGB1 and levels of Aβ42, Aβ40, and tau (Supplementary Fig. 3). Levels of pTau were not related to levels of CSF-HMGB1 in MCI patients (Supplementary Fig. 3). Moreover, we detected no relationship between CSF-HMGB1 and Aβ42, Aβ40, tau, or pTau in the AD group (Supplementary Fig. 3). The number of patients in which both Aβ42 and Aβ40 could be analyzed was small, so this result is not informative (Supplementary Fig. 3).

We observed no relationship between ApoE4 allele copy number and CSF-HMGB1 in the MCI group. However, ApoE4 was negatively correlated with CSF-HMGB1 in the AD group (Supplementary Fig. 3). This finding may be of interest, assuming that the summative pathology linked to CSF-HMGB1 and ApoE4 allele copy number reflects cognitive impairment.

**Necrosis occurs most actively in the MCI stage**. HMGB1 is a representative damage-associated molecular patterns (DAMPs) molecule released from necrotic cells[23,24]. Our findings in human CSF suggested that neuronal necrosis might occur more frequently in preclinical MCI than in symptomatic AD. Evaluation of cell death in vivo has been technically difficult because intensities of cell death markers diminish rapidly after cell death or are cleared by phagocytes in the brain. To overcome the difficulty, we employed anti-pSer46-MARCKS antibody whose reactivity was characterized by western blot[22]. The specificity to pSer46-MARCKS was further confirmed by ELISA using phospho- and non-phospho peptides matching to the 14 amino acid sequence of MARCKS around Ser46 (Supplementary Fig. 4a). Anti-pSer46-MARCKS antibody was purified with affinity columns of non-phosphorylated antigen peptide and the phosphorylated antigen peptide (Supplementary Fig. 4b). Therefore, we compared the reactivity of anti-pSer46-MARCKS antibody and anti-non-phosphorylated MARCKS antibody in immunohistochemistry of cerebral cortex from 5xFAD mice at 6 months (Supplementary Fig. 4c). Obviously the patterns were different, and anti-pSer46-MARCKS antibody, but not anti-non-phosphorylated MARCKS antibody, stained structures

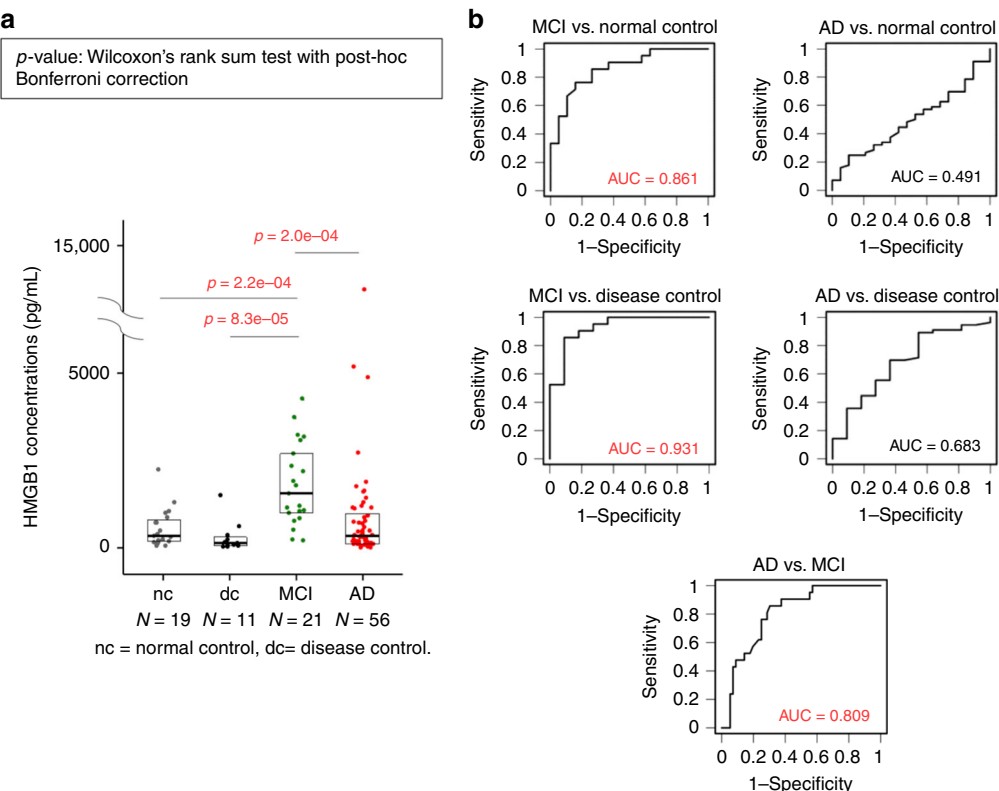

**Fig. 1 HMGB1 levels are elevated in the CSF of MCI and AD patients. a** CSF-HMGB1 levels in the normal control (nc) (N = 19 persons), disease control (dc) (N = 11 persons), MCI (N = 21 persons), and AD (N = 56 persons) groups were evaluated by high-sensitivity ELISA. The box plot shows the median and quartiles. Statistical differences among groups were evaluated using the Wilcoxon rank–sum test with post-hoc Bonferroni correction. **b** Receiver operating characteristic (ROC) curves for the MCI or AD group versus the normal control (nc) and disease control (dc) groups. Area under the ROC curve (AUC) values are shown in the graphs. Source data are provided as a "Source Data file".

around extracellular Aβ aggregates (Supplementary Fig. 4c). We also performed western blot to examine chronological change of pSer46-MARCKS in the cerebral cortex of 5xFAD mice from 1 to 12 months (Supplementary Fig. 4d). Interestingly, pSer46-MARCKS formed high molecular weight smear (HMW), suggesting that the character of MARCKS as an intrinsically disordered/denatured protein (IDP)[22,26] (Supplementary Fig. 4e) was enhanced by phosphorylation at pSer46. HMW smear, 80 kD and 50 kD bands were all increased during pathological progression (Supplementary Fig. 4d, right graph). In addition, pSer46-MARCKS was increased also during normal aging of non-transgenic sibling mice (Supplementary Fig. 4d, right graph). Consequently, the ratio of pSer46-MARCKS between 5xFAD mice and non-transgenic sibling mice were declined after 3 months, consistently with our previous result of the similar ratio in mass analysis[22].

pSer46-MARCKS reactivity increased in neurons surrounding dying cells, enabling us to detect active neuronal necrosis at the moment of dissection[22] (Fig. 2a). Neurons under such active necrosis were marked by deformed and/or shrinking nuclei, sometimes with faint DAPI staining surrounded by degenerative neurites reactive for pSer46-MARCKS (Fig. 2a). Consistent with this, immunoelectron microscopy (IEM) confirmed that degenerative neurites reactive to pSer46-MARCKS antibody and full of autophagosomes surrounded intracellular Aβ plaques (Supplementary Fig. 5a, b). Beyond the borders of degenerative neurites (Supplementary Fig. 5a, yellow dot lines), amyloid plaques included cytoplasmic organelles (Supplementary Fig. 5a, white arrows). Immunohistochemistry also revealed that the similar degenerative neurites surrounded non-apoptotic dying neurons

(no chromatin condensation) with a deforming and shrinking nucleus (Supplementary Fig. 5c). These findings indicated that neurons died by necrosis at the center of degenerative neurite clusters, Aβ persisting after cell death served as a seed for further extracellular amyloid aggregation, and that such necrotic neurons released DAMPs such as HMGB1, tau, and Aβ. Moreover, by using primary mouse cortical neurons, we confirmed that neurons under α-amanitin-induced necrosis[17] but not glutamate-induced apoptosis[27–29] induced reactive increase of pSer46-MARCKS in surviving neurons in neighborhood (Supplementary Fig. 6a). Western blot also supported induction of pSer46-MARCKS in neurons by α-amanitin-induced necrosis but not glutamate-induced apoptosis (Supplementary Fig. 6b). These findings further supported that reactive pSer46-MARCKS in neighboring cells could be used as a marker specifically indicating necrotic change of the central neuron which they surrounded.

In this work, we strictly defined "active necrosis" as a single dying cell surrounded by reactive pSer46-MARCKS signals (Fig. 2a). Since necrotic cells or apoptotic cells not removed by phagocytes are known to trigger secondary necrosis[30–32], we defined "secondary necrosis" as a cluster of multiple cells with reactive pSer46-MARCKS signals (Fig. 2a). Most extracellular Aβ aggregates in 5xFAD mice were associated with pSer46-MARCKS and DAPI signals. However, in aged mice, a small part of Aβ aggregates show disappearance or weakening of pSer46-MARCKS and DAPI signals, which we named as "ghost of cell death".

We found that the proportion of active necrosis increased during the preclinical stage of 5xFAD mice[33], from 1 to 6 months, and then decreased from 12 to 18 months after the onset of cognitive impairment (Fig. 2b). A similar relationship between

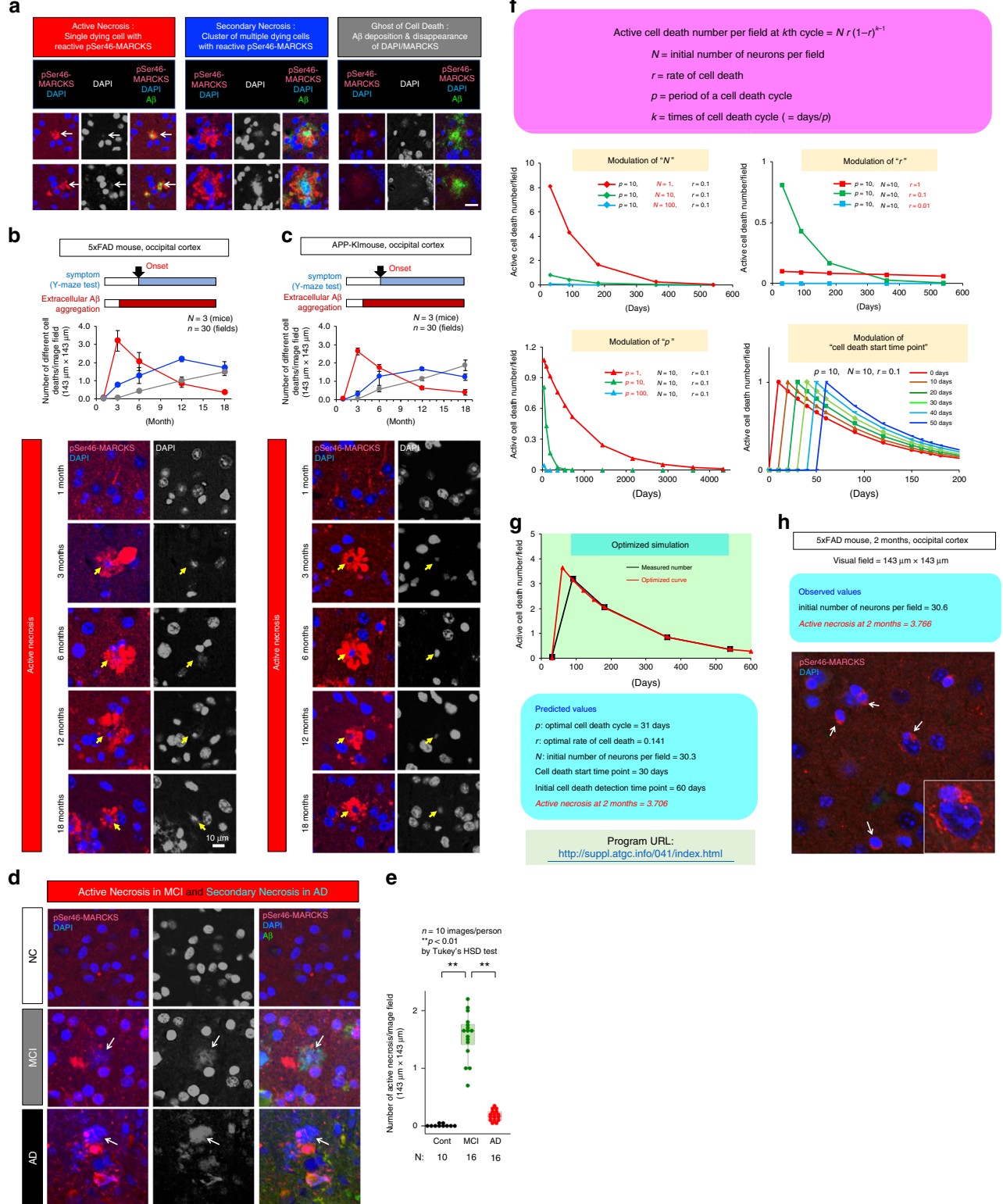

clinical stage and active necrosis was confirmed in human mutant APP knock-in mice (APP-KI mice)[34] (Fig. 2c). This finding that cell death precedes extracellular Aβ aggregate appearance at 3 months in 5xFAD mice[33] and at 4 months in APP-KI mice[34] was unexpected. However, the early-stage appearance of active necrosis in mouse models (Fig. 2b, c) explains the elevation in the CSF-HMGB1 level in human MCI (Fig. 1) and presumably agrees with results of previous clinical trials.

Immunostaining of pSer46-MARCKS with postmortem human brains of MCI and non-neurological disease control patients confirmed that cortical neurons underwent morphologically similar necrosis in MCI patient brains (Fig. 2d). The active necrosis revealed by pSer46-MARCKS were present at significantly higher frequencies in all MCI patients than in AD patients (Fig. 2d, e), and neurons surrounded with reactive pSer46-MARCKS, even though they did not match the strict criteria of

**Fig. 2 Necrosis occurs most actively at the preclinical stage in mouse and human AD brains. a** Morphological definition of active necrosis, secondary necrosis and ghost of cell death. Active necrosis is specified by a single degraded nucleus detected by DAPI surrounded by pSer46-MARCKS-positive degenerative neurites. Secondary necrosis is a cluster of multiple dying cells with residual Aβ in extracellular space and surrounding pSer46-MARCKS stains. Ghost of cell death is an extension of secondary necrosis in which DAPI and pSer46-MARCKS stains have faded out. **b** Upper graphs show time course of active necrosis, secondary necrosis and ghost of cell death in 5xFAD mice, in retrosplenial dysgranular cortex. Representative images from each time point are shown below the graph. Yellow arrow indicates a single degraded nucleus surrounded by reactive pSer46-MARCKS stains. $N = 3$ mice, $n = 30$ fields. **c** Time course of active necrosis in human mutant APP knock-in mice. Time course of necrosis ($N = 3$ mice, $n = 30$ fields) and representative images are shown. **d** Representative images of active necrosis in human MCI (MCI) and non-neurological disease control (NC). Rupturing or deformed nucleus undergoing necrosis is surrounded by Aβ and pSer46-MARCKS-positive degenerative neurites (white arrow). **e** The box plot shows the number of active necrosis per visual field in the median, quartiles and whiskers that represent 1.5× the interquartile range. **$p < 0.01$, Tukey's HSD test ($n = 10$ images/person). **f** Simulation of active necrosis. A formula was generated by assuming that cell death occurs at a constant rate in the residual neurons and in regular time interval (top). Modulation of each parameter changed simulation curves (graphs). **g** Numerical simulation program generated the optimized curve (red line) based on observed values of active necrosis in occipital cortex of 5xFAD mice (black line) and predicted parameter values and active necrosis at an unmeasured time point (2 months). **h** The number of active necrosis observed afterwards with samples at 2 months (60 days) matched exactly with the predicted number. Values in each group are summarized by mean ± S.E.M. Source data are provided as a "Source Data file".

active necrosis because the nuclear DAPI stain remained intact, were increased in MCI (Supplementary Fig. 7). The number of neurons by itself was decreased in AD in comparison to that in MCI (Supplementary Fig. 7), and neurons surrounded with reactive pSer46-MARCKS were remarkably decreased in symptomatic AD.

**Mathematical simulation of active necrosis.** The chronological change of active necrosis motivated us to mathematically simulate chronological changes of active necrosis (Fig. 2f). If total number of neurons at the initial time point is $N$, and cell death (active necrosis) occurs constantly at the rate of $r$, residual number of neurons at current time ($N_k$) is calculated as follows.

$$N_k = N(1 - r)^{k-1} \qquad (1)$$

Here, $k$ is the number of cell death cycles, and $k$ is calculated by the period necessary for a single turn of cell death and the time from the initial time point when cell death starts to the current time point.

Then, active cell death is calculated as following.

$$\text{Active cell death} = N_k - N_{k-1} = N\, r\, (1 - r)^{k-1} \qquad (2)$$

The simulation curve changed when the parameters, $N$, $r$, $p$ and the initial detection time point ($p$ days later than initiation time point of cell death) were modulated (Fig. 2f).

As the graph shows, chronological change of actually observed active necrosis was precisely simulated (Fig. 2g). The consistency between theoretical and experimental data was surprising. The parameter deduced from observed number of active necrosis suggested that cell death period is 31 days and cell death ratio is 0.141 (14.1% of cell death die in 31 days). Initial number of neurons (30.3 cells) matched exactly with the neuronal number actually observed (30.6 cells) (Fig. 2g).

In addition, the mathematical simulation predicted that active necrosis process initiates from 1 month when intracellular Aβ begins to be detected in immunohistochemistry[22] and it should reach to 3.706 cells per area (143 μm × 143 μm) at 2 months (Fig. 2g). Therefore, we examined again the brains of 5xFAD mice at 2 months, and surprisingly found that the actual frequency active necrosis (3.766 cells/area) matched exactly with the expected value (Fig. 2h). These consistencies in the mathematical induction and deduction further supported our theory for dynamics of active necrosis.

**ER enlargement is a morphological feature of necrosis in MCI.** To characterize necrosis in vivo, we employed two-photon microscopy[19] and analyzed dynamic changes of the ER in

cortical neurons of 5xFAD mice from 1 (pre-symptomatic/pre-clinical stage) to 6 months (symptomatic/clinical stage) (Fig. 3a, b). The ER and Aβ were visualized using ER-Tracker™ and BTA1, respectively. At 1 month, ER volume was larger and less stable in 5xFAD than in non-transgenic sibling mice (B6/SJL) (Fig. 3a, b), and this tendency persisted at later time points (Fig. 3b, Supplementary Fig. 8). Moreover, these mice had a higher standard deviation or quartile deviation of ER volume, indicating that the ER was unstable in 5xFAD mice from 1 to 6 months (Fig. 3c). After two-photon microscopy, the mouse brains were investigated by electron microscopy. ER enlargement was confirmed at high frequencies in neurons of 5xFAD mice but rarely in non-transgenic sibling mice (B6/SJL) (Fig. 4a).

We extended electron microscopic analysis to human brains of non-neurological disease, MCI and AD patients (Fig. 4b). Remarkable enlargement of ER equivalent to the finding in 5xFAD mice was detected in MCI at a higher frequency than in AD patients (Fig. 4b). Instead, the frequency of extracellular aggregates was increased in AD (Fig. 4b, asterisk). Larger magnification revealed ribosomes on ER membrane confirming the origin of the ballooned organelles (Fig. 4c). A few ribosomes remained on the surface of extremely enlarged vacuoles (Fig. 4b, arrows in #1 and #2 of MCI) indicating that they originated from rough ER. Consistently, immunohistochemistry with anti-MAP2 and anti-calnexin (ER membrane marker) antibodies or with anti-MAP2 and anti-KDEL (ER content marker) antibodies revealed ER enlargement in cortical neurons of MCI patients (Fig. 4d) and of pre-symptomatic 5xFAD mice (Fig. 4e).

**Intracellular Aβ deprives YAP from the nucleus.** The ER enlargement and instability we observed in 5xFAD mice were reminiscent of transcriptional repression-induced atypical cell death (TRIAD), the Hippo pathway-dependent necrosis[17–19]. Hence, we investigated key molecules in the Hippo pathway in human postmortem brains of MCI (amnestic MCI with AD pathology) and symptomatic AD patients. First, we discovered that intracellular Aβ aggregates deprived YAP from the nucleus, ultimately causing a decrease in nuclear YAP levels in the cortical neurons of AD and MCI patients (Fig. 5a). This remarkable finding was observed in three MCI (amnestic MCI with AD pathology) and three symptomatic AD patients (Fig. 5b, upper graph). In MCI, DAPI signal intensities were decreased in neurons where intracellular Aβ aggregates deprived YAP from the nucleus (Fig. 5b, middle graph). Comparison among control, MCI and AD also confirmed decrease of DAPI signal intensities in cortical neurons with cytoplasmic YAP/Aβ-colocalization (Fig. 5b, lower graph). Immunoprecipitation of cerebral cortex tissues (temporal lobe) from human AD patients who had been

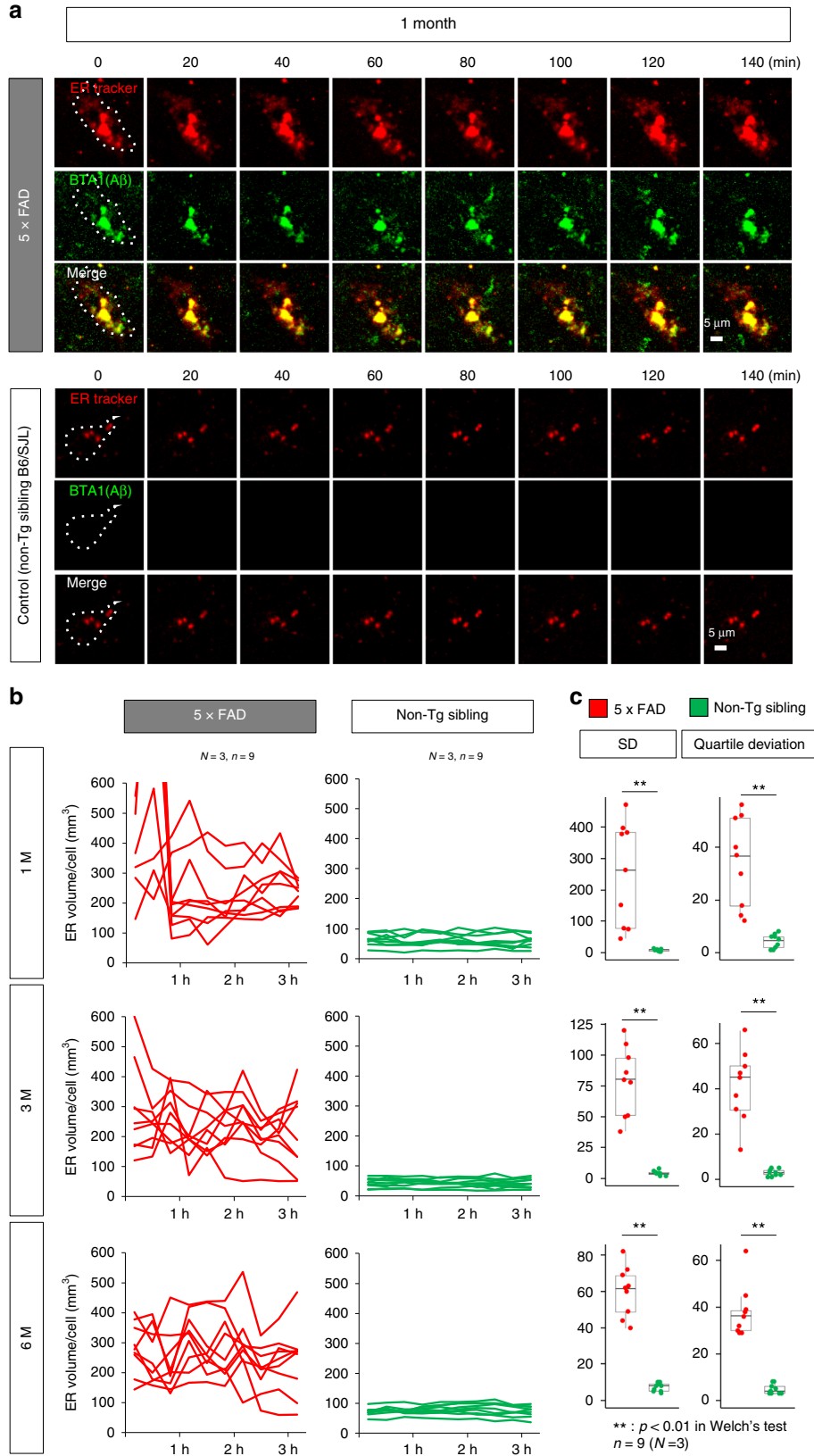

pathologically diagnosed as pure AD (Fig. 5c) also supported the interaction between YAP and Aβ. Consistent with the reduced nuclear YAP in immunohistochemistry, western blot revealed a similar decrease of YAP in temporal and occipital tip tissues from AD patients (Fig. 5d). YAP and Aβ levels were inversely correlated in cortical tissues from the occipital tip and temporal tip of AD patients and non-neurological disease controls (Fig. 5e).

In addition, LATS1 kinase, which prevents nuclear translocation of YAP[35], was activated in human cortical neurons of MCI and in those of AD to a lesser extent (Supplementary Fig. 9a, b).

**Fig. 3 Extreme instability of ER in AD model mice revealed by in vivo ER imaging. a** In vivo ER and Aβ images were acquired by two-photon microscopy from 1-month-old 5xFAD mice into which ER-tracker and BTA1 had been injected in one shot 4 h before observation. ER and Aβ image sets were taken in tandem every 20 min. 3D images of ER and Aβ stains were merged by IMARIS (Bitplane, Zurich, Switzerland). Dot-line indicates a single neuron. **b** Total volumes of ER puncta belonging to a single cell were quantified by IMARIS (Bitplane, Zurich, Switzerland), and time courses are shown in the graph. Changes were more pronounced in 5xFAD mice than in non-transgenic sibling mice (Non-Tg sibling). $N = 3$ mice, $n = 9$ cells. **c** To verify the finding in **b**, standard deviation (SD) and quartile deviation of ER volumes from a single cell at multiple time points were compared between groups of 5xFAD and non-transgenic sibling mice. Box plots show the median, quartiles and whiskers that represent 1.5× the interquartile range. P-values were determined by Welch's test, $^{**}p < 0.01$ ($N = 3$ mice, $n = 9$ cells). Source data are provided as a "Source Data file".

On the other hand, PLK1, which switches necrosis to apoptosis[19], was not activated in either MCI or symptomatic AD (Supplementary Fig. 10a, b).

The decrease of nuclear YAP due to cytoplasmic co-segregation with Aβ was confirmed in cortical neurons of 5xFAD mice (Supplementary Fig. 11a, b) and human mutant APP-KI mice (Supplementary Fig. 11c, d) at 3 months, prior to the onset of cognitive impairment. In these neurons DAPI signal intensities were decreased (Supplementary Fig. 11b, d). LATS1 activation in cortical neurons was also confirmed by immunohistochemistry in both mouse strains (Supplementary Fig. 9c, d). These findings support that Hippo pathway–dependent necrosis (TRIAD)[17–19] occurs from the pre-symptomatic to post-symptomatic stages in both human and mouse AD pathology.

Moreover, essential transducers of necroptosis, RIP1/3 and the downstream pathways were not activated in the pathway analysis based on comprehensive phosphoproteome data (Supplementary Fig. 12a), in western blot (Supplementary Fig. 12b) and immunohistochemistry (Supplementary Fig. 12c) of cerebral cortex tissues of 5xFAD mice from 1 to 3 months when YAP-dependent necrosis occurred at high frequencies.

These results further support that the necrosis at the early stage of AD pathology is distinct from necroptosis, which had been implicated in neuronal loss at the late stage after extracellular Aβ aggregation[36]. In human postmortem brain of MCI due to AD, RIP1/3 were not also activated in cortical neurons possessing intracellular Aβ (Supplementary Fig. 12d).

**YAP deprivation by intracellular Aβ induces Hippo pathway-dependent necrosis.** To further uncover the mechanism of intracellular Aβ induced-necrosis, we employed human induced pluripotent stem cells (iPSCs) carrying heterozygous or homozygous APP mutations (KM670/671NL) generated by genome editing[37], differentiated them into neurons, and performed time-lapse imaging to elucidate the chronological relationship among amount of intracellular Aβ, transcriptional activity of TEAD-YAP, and ER ballooning. ER ballooning and rupture in heterozygous and homozygous AD-iPSC–derived neurons occurred at a higher frequency than in normal iPSC-derived neurons (Fig. 6a, b, Supplementary Movies 1–3), consistently with the observation in vivo (Fig. 3, Supplementary Fig. 8) and TRIAD[17–19].

Interestingly, BTA1 signals reflecting intracellular Aβ were increased nearly 10 h before initiation of ER ballooning (Fig. 6c). In addition, a TEAD-reporter vector[38], which is composed of the TEAD-responsive element flanked to mCherry gene to monitor YAP co-transcriptional activity and the CMV-promoter flanked to EGFP to detect transfected cells (Fig. 6d), revealed that TEAD-YAP transcriptional activity was declined 8 h before ER ballooning in accordance with the increase of BTA1-stained intracellular Aβ (Fig. 6e).

We also confirmed that siRNA-mediated knockdown of YAP (Fig. 6f) directly induced ER ballooning in human normal iPSC-derived neurons (Fig. 6g). Since YAP-siRNA decreased YAP protein in immunocytochemistry (Fig. 6h) and western blot (Fig. 6i) at the time point of 0 min, the duration from the decrease

of TEAD/YAP transcriptional activity to the initiation of ER ballooning was estimated to be 2–4 h. YAP-siRNA significantly increased HMGB1 released from necrotic iPSC-derived neurons (Fig. 6j).

Injection of YAP-siRNA into cerebral cortex of normal control mice (B6/SJL) promptly induced ER instability of transfected cortical neurons under in vivo imaging by two-photon microscopy (Fig. 7a, b). Knockdown of YAP protein in siRNA-transfected neurons was confirmed in immunohistochemistry (Fig. 7c) and western blot analyses of cortex tissues (Fig. 7d). Intriguingly, we found patchy stains of pSer46-MARCKS (Fig. 7e) induced by HMGB1, a DAMPs molecule released from necrotic cells[23,24]. A high magnification of such a patchy stain revealed a single or few YAP-siRNA-transfected cells with extremely weak stains of DAPI surrounded by pSer46-MARCKS (Fig. 7e) that matched well with the criteria of nuclear morphology to define active necrosis. Decreased nuclear volume revealed by quantitative analysis with 30 μm sections of cortex tissues after YAP-knockdown also supported TRIAD necrosis (Fig. 7f).

Moreover, we observed the whole processes from Aβ accumulation to ER ballooning via repression of TEAD-YAP transcriptional activity in a single iPSC-derived neuron with heterozygous or homozygous AD mutations (APP KM670/671NL) (Fig. 8a). EGFP-YAPdeltaC61, the neuronal isoform YAP that has the similar dynamics and roles to full-length YAP in TRIAD[19], was electroporated into neurospheres and differentiated into neurons. During the process, EGFP-YAPdeltaC61 was co-segregated to cytoplasmic Aβ (Fig. 8b, magenta arrow), and deprived from the nucleus (Fig. 8b). On ER ballooning, YAP was further shifted to the ER ballooning protrusion (Fig. 8b, green arrow) and released by rupture (Fig. 8b, white arrow), while cytoplasmic Aβ remained as aggregates (Fig. 8b, blue arrow). All the processes are also shown in movie (Supplementary Movies 4–6). We quantitatively confirmed in each neuron that the increase of BTA1 signal intensity was followed by the decrease of YAPdeltaC in the nucleus (Fig. 8c, Supplementary Movies 4–6).

Moreover, immunohistochemistry with anti-YAP and anti-calnexin antibodies revealed ER enlargement in YAP-deficient neurons of human MCI patients (Fig. 8d). Consistently with iPSC-derived neuron carrying APP mutations (Fig. 8b), YAP was aggregated in the cytoplasm or translocated into ER ballooning (Fig. 8d). The similar ER ballooning was also observed in postmortem human brains of AD patients but at a lower frequency (Fig. 8d).

Timelapse imaging by two-photon microscopy revealed that a small part of neurons possessing intracellular Aβ underwent TRIAD necrosis and the residual intracellular Aβ after neuronal rupture might become seed for extracellular Aβ aggregation (Supplementary Fig. 13). Since observation of the ER rupture in vivo was far more difficult technically, we could not detect the whole processes in a single neuron in vivo. However, these data in vivo and in vitro collectively suggested the sequential pathological processes of intracellular accumulation of Aβ, deprivation of YAP from the nucleus linked with suppression of TEAD-YAP transcriptional activity, and ER ballooning.

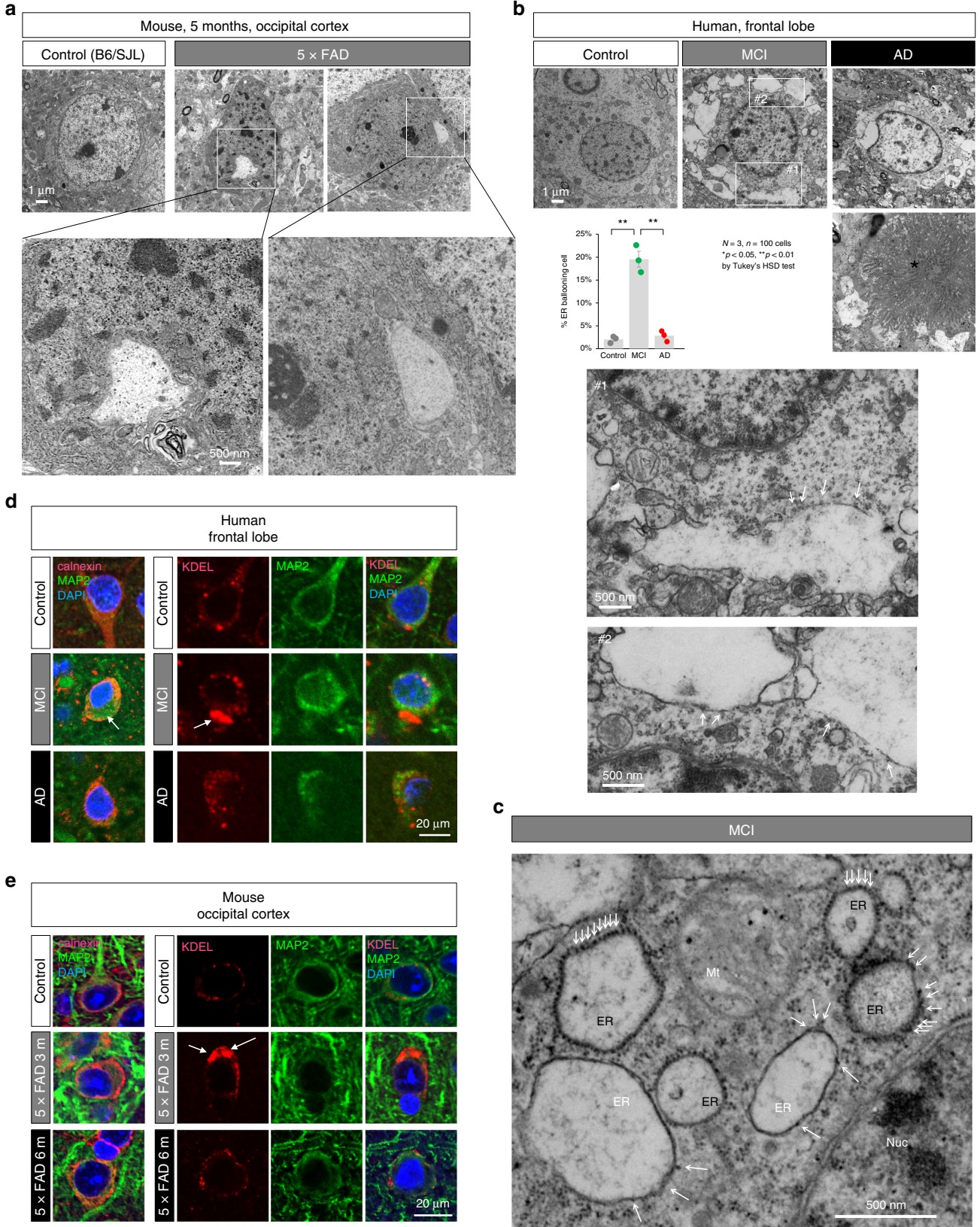

**S1P and YAPdeltaC rescue ER instability, necrosis, and cognitive impairment in vivo**. Next, we investigated whether sphingosine-1-phosphate (S1P) and YAPdeltaC61, a neuron-specific isoform of YAP possessing a similar rescue effect to that of full-length YAP on Hippo pathway–dependent necrosis[17–19],

rescue the pathology of 5xFAD mice. Regarding S1P, continuous intrathecal administration (40 nM, 0.15 μL/h) into the CSF space was initiated at either 1 or 5 months and continued until 6 months (Fig. 9a). Regarding AAV-YAPdeltaC, one-shot injection ($1 \times 10^{10}$ vector genomes/mL × 1 μL) into the CSF space

**Fig. 4 ER enlargement of neurons in 5xFAD mice and human MCI/AD patients. a** Electron microscopy of neurons in control (B6/SJL) and 5xFAD mice. Marked ER enlargement was observed in 5xFAD mice at 5 months of age before the onset. **b** Electron microscopy of neurons in non-neurological disease control, MCI (Braak stage III by Gallyas-Braak staining) and AD (Braak stage V) patients. Higher magnification in two subcellular fields reveal ribosomes attached to vacuoles (arrow). Amyloid plaque in AD patient is shown (asterisk). Quantitative analysis of three groups is shown in the graph. The bar graph indicates average and mean ± S.E.M., together with the corresponding data points. P-values were determined by Tukey's HSD test, **$p < 0.01$ ($N = 3$ persons, $n = 100$ cells). **c** A high magnification image of ballooned ER. Ribosomes on ER membrane help identification of the origin of the medium-size ERs (black ER), while ribosomes were detached when ER lumen was further enlarged and only a few ribosomes were remained (white ER). Mt: mitochondria. Nuc: nucleus. **d** Calnexin and MAP2 or KDEL and MAP2 co-staining of human neurons of non-neurological disease (control), MCI and AD patients. Remarkable enlargement of ER was detected (arrow) in MCI more frequently than in AD. **e** Calnexin and MAP2 or KDEL and MAP2 co-staining of mouse neurons of 5xFAD mice at 3 and 6 months of age. Remarkable enlargement of ER was detected (arrow) at 3 months more frequently than at 6 months. Source data are provided as a "Source Data file".

between the dura and brain parenchyma was performed at 1 or 5 months, and the same series of examinations was performed (Fig. 9a). When administered from or at 1 month, S1P and AAV-YAPdeltaC restored the alteration rate in the Y-maze test in 5xFAD mice (Fig. 9b), although their therapeutic effects were somewhat smaller by administration from or at 5 months (Fig. 9b).

Consistently, two-photon microscopy revealed stabilization of ER volume in 5xFAD mice by S1P and AAV-YAPdeltaC (Fig. 9c). Immunohistochemistry revealed that S1P and AAV-YAPdeltaC decreased the extracellular Aβ burden (Fig. 9d) in addition to the increase of nuclear YAP/YAPdeltaC (Fig. 9d). The decrease in the abundance of extracellular Aβ plaques (Fig. 9d) was further confirmed by western blot (Fig. 9e) and ELISA (Fig. 9f). YAPdeltaC and total YAP were increased after the S1P and AAV-YAPdeltaC treatments in cortex tissues by western blot (Fig. 9g) and in cortical neurons by immunohistochemistry (Fig. 9h). The decrease of extracellular Aβ aggregation could be explained by assuming that intracellular Aβ accumulation serves as a seed for extracellular Aβ aggregation after cell death[22,39,40]. However, further investigation is necessary to elucidate the relationship between intracellular Aβ accumulation and extracellular Aβ aggregation, as well as the relationship between Aβ metabolism and the Hippo pathway. Given that S1P and AAV-YAPdeltaC inhibit necrosis by increasing the effector molecule YAP, it is clear why intracellular Aβ levels were unchanged despite a reduction in necrosis (Fig. 9d). Normal sibling mice (B6/SJL) after the similar treatments of S1P and AAV-YAPdeltaC were also examined for YAP expression, intracellular Aβ and extracellular Aβ levels (Supplementary Fig. 14).

**S1P and YAPdeltaC rescue ER instability in AD-iPS-cell–derived neurons**. To further evaluate S1P and AAV-YAPdeltaC as a candidate therapeutic strategy in human AD, we employed human iPSCs-derived neurons carrying heterozygous or homozygous AD mutations (APP KM670/671NL) introduced by genome editing (Fig. 10a). As mentioned in previous experiments (Fig. 6), we detected ER ballooning and rupture occurred in heterozygous and homozygous AD-iPSC–derived neurons (Fig. 10b; Supplementary Movies 1–3). BTA1 barely stained normal iPSC–derived neurons, but stained >75% of AD-iPSC–derived neurons, reflecting intracellular Aβ accumulation (Fig. 10b, c). The frequency of ER ballooning and resultant cell death identical to the TRIAD reported in Huntington's disease[17] were obviously higher in non-treated AD-iPSC–derived neurons than in non-treated normal iPSC-derived neurons (Fig. 10b, d). AD-iPSC–derived neurons accumulating intracellular Aβ underwent TRIAD at a higher frequency as aforementioned (Fig. 10d, e; Supplementary Movies 7–9). As expected, 20 nM S1P did not affect intracellular Aβ accumulation (Fig. 10c) but significantly suppressed the frequency of ER ballooning and resultant cell death

both in total neurons and in neurons with intracellular Aβ accumulation (Fig. 10d, e; Supplementary Movies 10–12).

Similarly, we tested the effect of AAV-YAPdeltaC on ER ballooning (Fig. 10f). In this independent experiment, BTA1 stained >75% of AD-iPSC–derived neurons (Fig. 10g). AAV-YAPdeltaC remarkably suppressed the frequency of ER ballooning and resultant cell death in AD-iPSC–derived neurons (Fig. 10g, h, i, j; Supplementary Movies 13–18). We also confirmed that S1P increased the level of nuclear YAP protein (Supplementary Fig. 15a), and that AAV-YAPdeltaC increased the level of nuclear YAPdeltaC protein (Supplementary Fig. 15b, c) in AD-iPSC–derived neurons. Interestingly, this overexpression of YAPdeltaC also restored nuclear endogenous full-length YAP (Supplementary Fig. 15d), presumably because overexpressed YAPdeltaC occupied intracellular Aβ, enabling endogenous YAP to undergo nuclear translocation.

Consistent with this, TEAD-YAP/YAPdeltaC–mediated transcription, which was suppressed in heterozygous and homozygous AD-iPSC–derived neurons due to sequestration of YAP into intracellular Aβ aggregates, was rescued by S1P or AAV-YAPdeltaC, as evaluated by luciferase assay using a TEAD-responsive element reporter plasmid (Fig. 10k, l). Meanwhile, S1P and YAPdeltaC did not affect the amount of intracellular Aβ, as determined by BTA1 (Fig. 10c, h) or anti-Aβ antibody (Supplementary Fig. 15e, f), supporting that intracellular Aβ accumulation occurs upstream of TEAD-YAP/YAPdeltaC–mediated transcription, ER instability and cell death.

BTA1-mediated amyloid labeling suggested that Aβ was mostly localized to the ER in AD-iPSC–derived neurons (Figs. 6, 10), consistent with the scenario outlined above. Higher magnification of BTA1-stained live neurons revealed that a small portion of Aβ shifted from ER to cytosol (green arrow, Supplementary Fig. 16), which could be developed to intracellular Aβ aggregates (Figs. 6, 8, 10). Interestingly, before necrosis processes initiate, a very small part of Aβ appeared to be excreted from cells as a small vesicle at several parts of the cell membrane (white arrow, Supplementary Fig. 16). Although the ER signals were weak in Z-stack images, ER components were sometimes co-located at such vesicles in single-slice confocal microscopy images (white arrow, Supplementary Fig. 16). Such Aβ secretion was also detected by immunocytochemistry with anti-Aβ antibody after fixation (Supplementary Fig. 15e, f, right panels). These results suggested that Aβ is secreted from intracellular Aβ–accumulating neurons by the exosome pathway via multivesicular bodies (MVBs).

**Other types of cell death in mouse AD models and human AD patients**. Finally, we summarize our data about co-existence of other types of cell death in the brains of AD model mice and human MCI/AD patients. Though a previous paper[36] suggested increased necroptosis in postmortem human AD brains, they used antibodies against non-phosphorylated RIP1/3 and did not show co-activation of RIP1/3 and MLKL. It is not

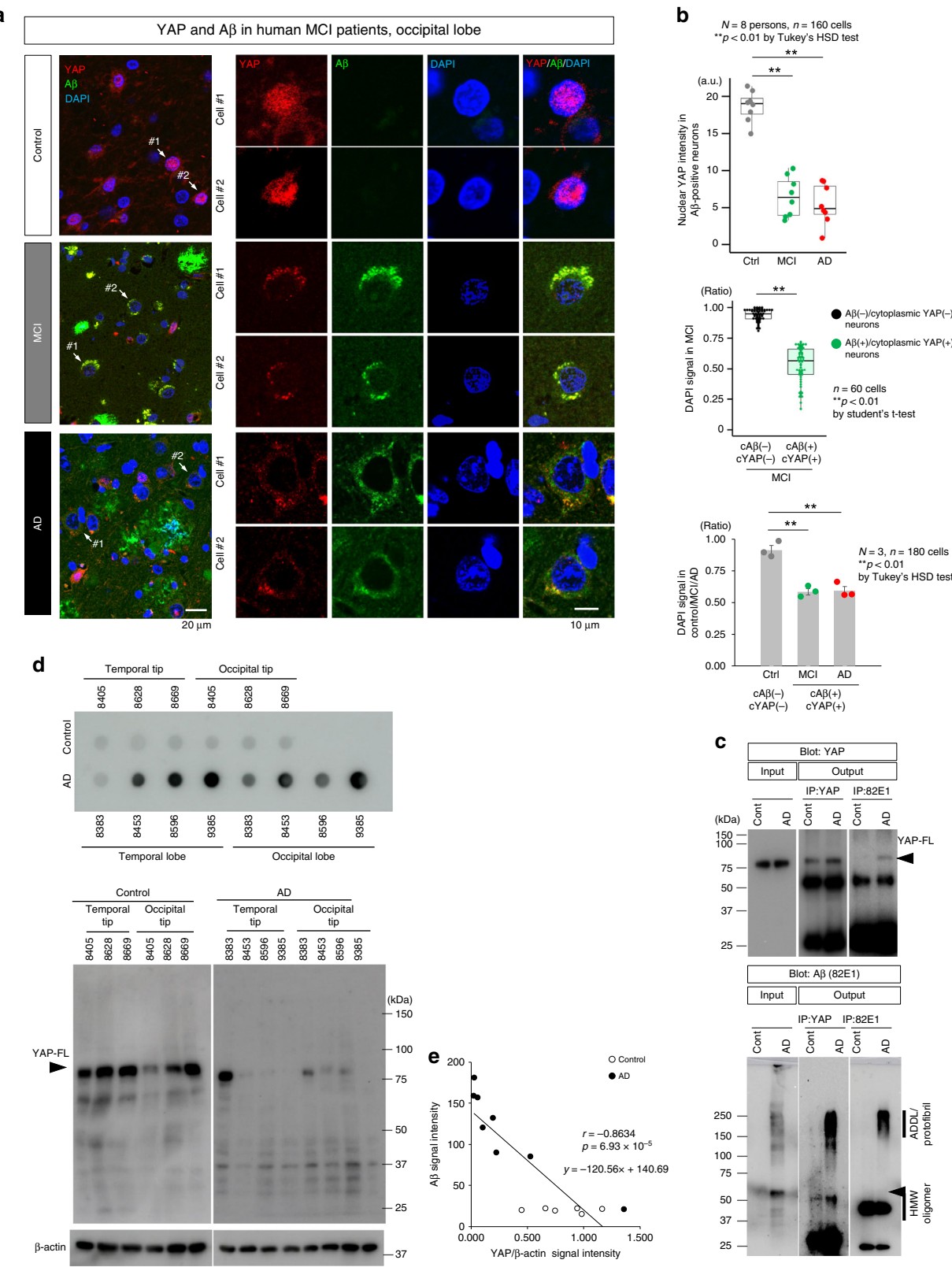

non-phosphorylated RIP1/3 and non-phosphorylated MLKL but phosphorylated RIP1/3 and phosphorylated MLKL that forms the signal transducing complex necrosome executing necroptosis[41–44]. In our immunohistochemistry of pRIP1/3 and pMLKL, with brain samples of AD model mouse and human AD patient, we could not detect co-localization of phosphorylated RIP1/3 and phosphorylated MLKL, which is essential for signal transduction of necroptosis, in any single neuron of 5xFAD or APP-KI mice at 3 months of age or of human MCI and AD patient (Supplementary Fig. 17). In parallel experiments of cerebral cortex tissues

**Fig. 5 Aβ sequesters YAP from the nucleus to intracellular aggregates. a** Immunohistochemistry of YAP and Aβ of human postmortem brains revealed sequestration of YAP into cytoplasmic Aβ aggregates and the resultant decrease of nuclear YAP in MCI and AD patients. **b** Quantification of signal intensities of nuclear YAP staining in neurons confirmed similar findings in three MCI patients and three symptomatic AD patients. A total of 160 neurons ($N = 8$ persons) in the occipital lobe were selected at random from each patient, and nuclear YAP signal intensities were quantified by confocal microscopy (FV1200IXGP44, Olympus, Tokyo, Japan). Signal intensity of DAPI per a nucleus was quantified in cortical neurons of occipital lobe of non-neurological disease controls and MCI or AD patients. The DAPI signals were compared in MCI between cytoplasmic YAP-positive and Aβ-positive neurons ($n = 60$) and cytoplasmic YAP-negative and Aβ-negative neurons ($n = 60$). In addition, signal intensity of DAPI per a nucleus of cytoplasmic was compared between YAP-positive and Aβ-positive neurons ($n = 180$) of MCI or AD patients ($N = 3$) and normal neurons ($n = 180$) of non-neurological disease controls ($N = 3$). Box plots show the median, quartiles and whiskers that represent 1.5× the interquartile range. The bar graph indicates average and mean ± S.E.M., together with the corresponding data points. P-values were determined by Tukey's HSD test or student's t-test, \*\*$p < 0.01$ **c** Immunoprecipitation reveals the interaction between YAP and Aβ in human postmortem brain. Upper panels: anti-Aβ antibody (82E1) co-precipitated YAP from cerebral cortex tissues of AD patients, but not from non-neurological disease control. Lower panels: reverse co-precipitation of Aβ protofibrils with YAP. **d** Western and dot blots of Aβ and YAP. Temporal tip and occipital tip tissues from pathologically diagnosed AD patients or controls were immunoblotted with anti-Aβ antibody (82E1, dot blot) or anti-YAP antibody (sc-15407). **e** Inverse correlation between Aβ burden and YAP in human patient/control brains. P-values were determined by Pearson's correlation coefficient (AD: $N = 8$ persons, Control: $N = 6$ persons). Source data are provided as a "Source Data file".

after ischemia as a positive control of necroptosis, co-localization of pRIP1/3 and pMLKL was confirmed almost in all neurons (Supplementary Fig. 17).

In our human AD-iPSC–derived neurons, morphological classification according to a previous report[19] revealed that <20% of neurons shrunk without cytoplasmic ballooning, mimicking apoptosis (Fig. 10m). However, the percentage of such apoptotic shrinkage or necrotic rupture was not significantly different among normal, heterozygous and homozygous APP-mutant neurons (Fig. 10n). Moreover, YAP-siRNA increased the shrinkage type of cell death while the extent of increase was not different between scrambled control siRNA and YAP-siRNA (Fig. 10o).

Collectively, these data supported that YAP-dependent TRIAD necrosis is a dominant form of cell death at the early stage of AD pathology, and could be a therapeutic target to cease the progression.

## Discussion

Morphological detection of neurons under the dying process is difficult because the cells lose both chemical and immunohistological staining. However, a sensitive marker (pSer46-MARCKS) of degenerative neurites surrounding dying neurons enabled us to detect necrosis efficiently. This technique revealed that the frequency of necrosis reaches a peak during the preclinical stage of AD pathology in two types of AD mouse models. Moreover, the technique revealed that active necrosis is more abundant at the prodromal stage of MCI than the clinical stage of AD in human patients (Fig. 2d, e). To the best of our knowledge, only the Herrup group has investigated cell death in MCI by using cell cycle markers while their focus was other than the chronological change of cell death[45].

Regarding the dynamics of active necrosis, we generated a formula based on the hypothesis that cell death occurs at a constant rate in the residual neurons and in regular time interval. Predicted number of active necrosis declined immediately after the onset of cell death, and explained well the actual chronological change (Fig. 2f, g). Moreover, multiple expected parameters also matched very well with the observed data (Fig. 2h) verifying the formula.

In addition, we determined that neuronal cell death in the early stage of AD is Hippo pathway-dependent necrosis, similar to that induced by RNA polymerase II inhibitor[17,18] or YAP sequestration by mutant Htt[19]. In the case of AD, YAP is sequestered to cytoplasmic Aβ, eventually impairing the function of YAP in the nucleus (Figs. 5–8). It remains unclear why YAP interacts with multiple causative proteins of neurodegenerative diseases.

However, YAP is a member of IDPs[46], a family that includes TDP43, FUS, tau, α-synuclein, and so on, which mutually interact and are involved in neurodegenerative diseases[47]. We found that low-complexity sequences are distributed throughout mouse and human YAP (Supplementary Fig. 18), supporting that full-length YAP and YAPdeltaC could interact with Aβ via intrinsically denatured regions.

Interestingly, a recent study implicated YAP as a hub molecule in AD pathology[10]. Xu and colleagues performed a meta-analysis of functional genomic data of AD and concluded that YAP is the most important hub molecule in the molecular network of AD[10]. Their subsequent experiments showed that YAP-KD increased the levels of Aβ[10], consistent with our results. Thus, the increase in the YAP mRNA level that they observed[10] could represent a protective transcriptional response aimed at compensating for the reduced level of YAP protein. Although their results did not reveal the direct relevance of YAP to neuronal cell death in AD, their findings match very closely with our observations and hypothesis.

Although cell death has been generally suspected as a terminal-stage pathology in AD, the evidence in support of this idea remains weak. Our experimental results suggest an alternative view regarding the timing and roles of cell death in AD (Supplementary Fig. 19). Intriguingly, Hippo pathway-dependent TRIAD necrosis occurs at an early stage and plays some critical roles in the progression of AD pathology. First, as a cell-autonomous process of degeneration, intracellular Aβ-induced necrosis decreases the number of cerebral neurons via sequestration of YAP. Second, as a non–cell-autonomous process necrotic neurons release alarmins/DAMPs that trigger secondary cell damage in surrounding neurons. This process could expand degeneration to bystander neurons that contain only a low level of intracellular Aβ. Third, after necrosis, intracellular Aβ becomes the seed for extracellular Aβ aggregation, representing another non-cell-autonomous means of expanding degeneration. Fourth, prionoid transmission of Aβ and tau proteins could be also promoted by TRIAD necrosis, as shown by live images of AD-iPSC-derived neurons (Supplementary Figs. 15, 16).

Restoration of the YAP protein level using an AAV vector successfully inhibited necrosis during the early stage of AD. More importantly, the treatment efficiently prevented cognitive impairment and extracellular Aβ aggregation in AD model mice. Paired experiments using AD-iPSC-derived neurons further supported the therapeutic effects of YAPdeltaC. Given that no extracellular Aβ aggregates existed under our culture condition, the experiment directly indicated that the necrosis was not

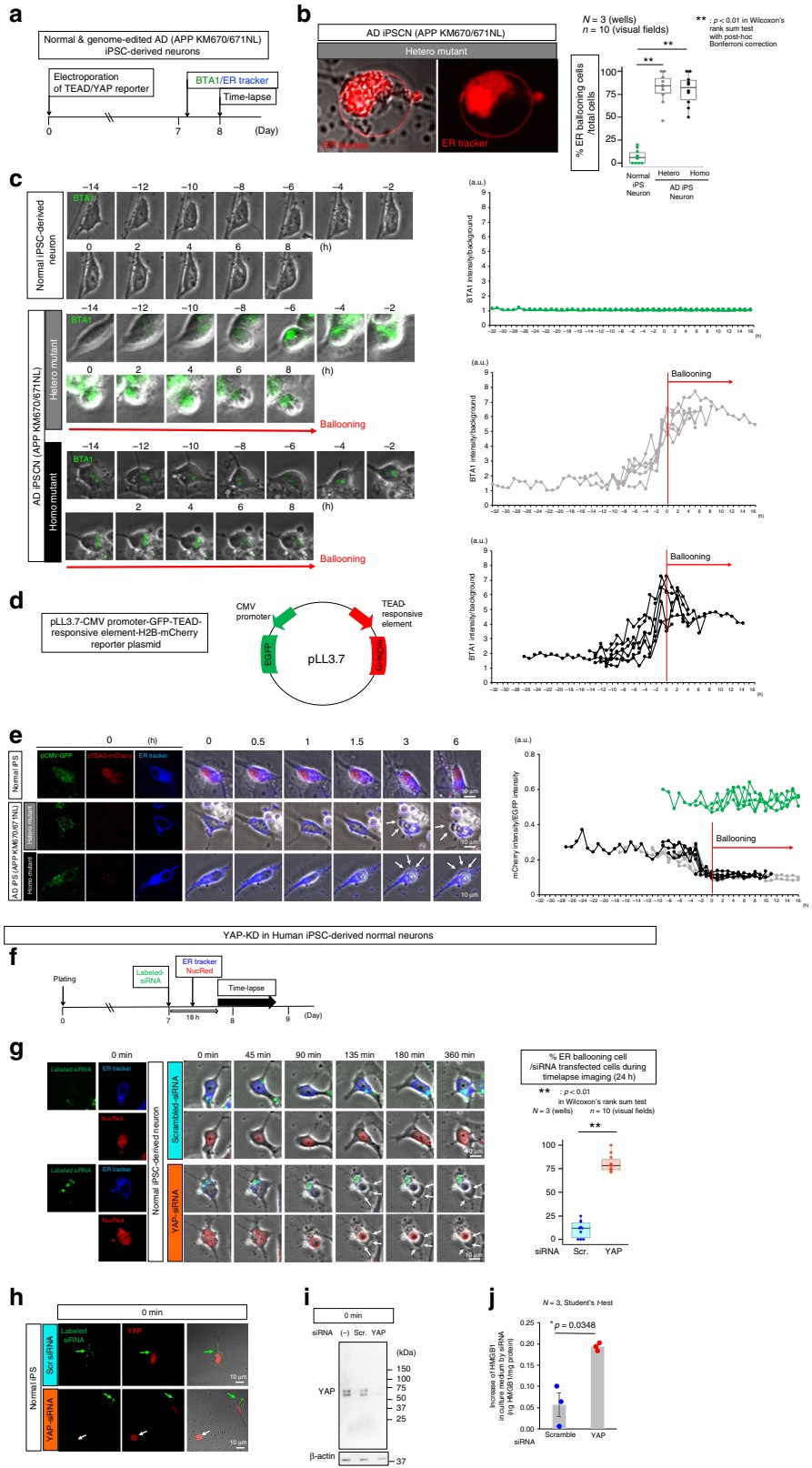

derived from extracellular Aβ aggregation but from intracellular Aβ accumulation. Early-stage intervention in molecules regulating Hippo pathway-dependent necrosis, or in triggering of necrosis by intracellular Aβ, could suppress progression to the late-stage pathological changes, possibly including extracellular

Aβ aggregation. Long-term follow-up of AAV-YAPdeltaC-treated mice for up to 6 months did not reveal tumors in systemic organs or the brain. However, further investigation at the GMP level vector would be necessary to finally confirm the safety of AAV-YAPdeltaC as a human therapeutic vector.

**Fig. 6 Time lapse imaging of multiple neurons suggests pathological cascade. a** Experimental protocol to evaluate the relationship among intracellular Aβ, TEAD/YAP transcriptional activity and ER ballooning. **b** High magnification revealed that the membrane of the cytoplasmic balloon was reactive to ER-tracker in iPSC-derived neurons carrying APP mutations (left panels). ER ballooning occurs frequently in iPSC-derived neurons carrying APP mutations (heterozygous and homozygous mutants carrying APP KM670/671NL) (right graph). *P*-values were determined by Wilcoxon's rank sum test with post-hoc Bonferroni correction, **$p < 0.01$ ($N = 3$ wells, $n = 10$ visual fields). **c** Accumulation of BTA1-stained intracellular Aβ occurs before ER ballooning in iPSC-derived neurons carrying APP mutations (left panels). Alignment of BTA1 signals changes in multiple neurons to the time point of ER ballooning initiation revealed intracellular Aβ began to accumulate 10 h before ER ballooning (right graphs). **d** Construction of a plasmid vector to monitor TEAD/YAP-transcriptional activity. **e** TEAD/YAP transcriptional activity was decreased in iPSC-derived neurons carrying APP mutations in comparison to normal iPSC-derived neurons (left panels). Alignment to ER ballooning time point revealed TEAD/YAP transcriptional activity started to decrease 8 h before ER ballooning. These results suggest a pathological cascade, increase of intracellular Aβ → decrease of TEAD/YAP transcriptional activity → ER ballooning. **f** Protocol of YAP-knockdown in normal human iPSC-derived neurons. **g** Time lapse imaging of siRNA-transfected neurons revealed the increase of ER ballooning by YAP-siRNA. Right graph shows quantitative analysis. P-values were determined by Wilcoxon's rank sum test, **$p < 0.01$ ($N = 3$ wells, $n = 10$ visual fields). **h** Conformation of siRNA-mediated YAP-knockdown by immunohistochemistry. Green and white arrows indicate siRNA-mediated and non-transfected cells, respectively. **i** Conformation of siRNA-mediated YAP-knockdown by western blot. **j** HMGB1 concentration in culture medium was quantified by using ELISA, and the increase of HMGB1 from the initial concentration after siRNA transfection was compared between scrambled control siRNA and YAP-siRNA. *P*-values were determined by Student's *t*-test ($N = 3$ wells). The box plot shows the median, quartiles and whiskers that represent 1.5× the interquartile range. The bar graph indicates average and mean ± S.E.M., together with the corresponding data points. Source data are provided as a "Source Data file".

Type III cell death with cytoplasmic changes in human AD brains that is homologous to TRIAD has been repetitively described in old historical papers of neuropathology. For instance, Hirano and colleagues[48] reported granulovacuolar body, which is a homologous large vacuole found in pyramidal neurons in Sommer's sector of senile dementia, AD and Pick's disease (now a form of FTLD). Another examples is the paper by Dickson and colleagues[49], which described ballooned neurons in AD, Pick's disease, cortico-nigral degeneration, and pigment-spheroid degeneration. Our previous work revealed TRIAD in Huntington's disease pathology[50], a number of papers have reported a cell death morphologically homologous to TRIAD[51–56].

The current definition of MCI is largely based on subjective complaints by patients who have insufficient cognitive decline to be diagnosed with dementia and who remain adequately socially adjusted. No objective markers are available to support the subjective diagnosis or to evaluate the pathological state during MCI stage. Therefore, in combination with amyloid PET to quantify the extracellular Aβ burden, the use of CSF-HMGB1 to detect the amount of on-going cell death could serve as a sensitive quantitative marker for evaluating disease progression and also the effect of candidate drugs.

In conclusion, we have provided evidence that neuronal necrosis induced by YAP deprivation occurs most actively in the early stages of AD, including preclinical AD, MCI or ultra-early stage of AD before extracellular Aβ aggregation. In addition, we showed that CSF-HMGB1 is a powerful tool for evaluating the activity of cell death in such stages. We also proposed therapeutic approaches targeting the change in the level of nuclear YAP in neurons, i.e., targeting the Hippo pathway–dependent necrosis.

## Methods

**Patient cohort.** A summary of all patient information is provided in Supplementary Table 1. Cohort 1 consists of four normal controls, one patient without dementia but with another neurological disease (disease control), 19 patients with MCI, and 18 patients with AD. Cohort 2 comprised 13 disease controls, seven MCI patients, and 17 AD patients. Cohort 3 comprised 30 normal controls and 30 AD patients. Cohort 4 comprised eight AD patients. Informed consent for the use of all human CSF was obtained and approved by the appropriate ethics committee at each institution and by Tokyo Medical and Dental University.

**Mini-Mental State Examination.** The Japanese version of the Mini-Mental State Examination (MMSE) was performed by the corresponding physician of each patient.

**CSF sampling.** All CSF samples were obtained by lumbar puncture before meal times and collected into polypropylene tubes. The CSF samples were centrifuged

($1000 \times g$ for 10 min at 4 °C) to remove any debris, and then stored in small aliquots at −80 °C.

**Aβ and tau measurement.** CSF-Aβ1–40 and -Aβ1–42 were measured by enzyme-linked immunosorbent assay (ELISA) using a human β amyloid (1–40) ELISA kit (292-62301, Wako Chemical Co., Saitama, Japan) and human β amyloid (1–42) ELISA kit (298-62401, Wako Chemical Co., Saitama, Japan). CSF-pTau proteins were measured using INNOTEST Phospho-tau (181 P, Innogenetics, Ghent, Belgium).

**High-sensitivity HMGB1 ELISA.** Polystyrene microtiter plates (152038, Nunc, Roskilde, Denmark) were coated with 100 μL of anti-human HMGB1 monoclonal antibody (1 mg/L, Shino-Test, Tokyo, Japan) in PBS and incubated overnight at 2–8 °C. The plates were washed three times with PBS containing 0.05% Tween 20, and then incubated for 2 h with 400 μL/well PBS containing 1% BSA to block remaining binding sites. After the plates were washed again, 100 μL of each dilution of the calibrator and CSF samples (1:1 dilutions in 0.2 M Tris pH 8.5, 0.15 M NaCl containing 1% BSA) was added to the wells. The plates were then incubated for 24 h at 37 °C. The plates were washed again, and then incubated with 100 μL/well peroxidase-conjugated anti-human HMGB1,2 monoclonal antibody (Shino-Test, Tokyo, Japan) for 2 h at room temperature. After another washing step, the chromogenic substrate 3,3′,5,5′-tetra-methylbenzidine (T022, Dojindo Laboratories, Kumamoto, Japan) was added to each well. The reaction was terminated with 0.35 M Na$_2$SO$_4$, and absorbance at 450 nm was read on a microplate reader (Model 680, Bio-Rad Laboratories, Hercules, CA, USA). A standard curve was obtained using purified pig thymus HMGB1 (Shino-Test, Tokyo, Japan). CSF samples with HMGB1 concentrations of 300 and 1000 pg/mL were analyzed to assess intra-assay ($n = 10$) and inter-assay precision ($n = 10$). The coefficients of variation in the intra- and inter-assay were 4.8–6.1% and 4.1–9.1%, respectively. The working range for the assay was 100–5000 pg/mL. Recovery of purified pig thymus HMGB1 added to pooled CSF was 80–105% ($n = 10$).

**Human tissue samples.** Paraffin sections and frozen brain tissues were prepared from human MCI/AD brains and disease control brains without dementia (non-neurological disease controls). Informed consent for the use of human tissue samples was obtained, after approval of the ethics committee at each institution and Tokyo Medical and Dental University.

**AD model mice.** 5xFAD transgenic mice overexpressing mutant human APP (770) with the Swedish (KM670/671NL), Florida (I716V), and London (V717I) familial Alzheimer's disease (FAD) mutations and human PS1 with FAD mutations (M146L and L285V) were purchased from The Jackson Laboratory (34840-JAX, Bar Harbor, ME, USA). Both the APP and PS1 transgenes were under the control of the mouse *Thy1* promoter[33]. The backgrounds of the mice were C57BL/SJL, which was produced by crossbreeding C57BL/6 J female and SJL/J male mice. APP-KI mice possess a single human APP gene with the Swedish (KM670/671NL), Arctic (E693G), and Beyreuther/Iberian (I716F) mutations[34].

**Behavioral analysis.** Exploratory behavior was assessed using a Y-shape maze consisting of three identical arms with equal angles between each arm (YM-3002, O'HARA & Co., Ltd., Tokyo, Japan). Mice at the age of 2 months were placed at the end of one arm and allowed to move freely through the maze during an 8 min session. The percentage of spontaneous alterations (indicated as an alteration

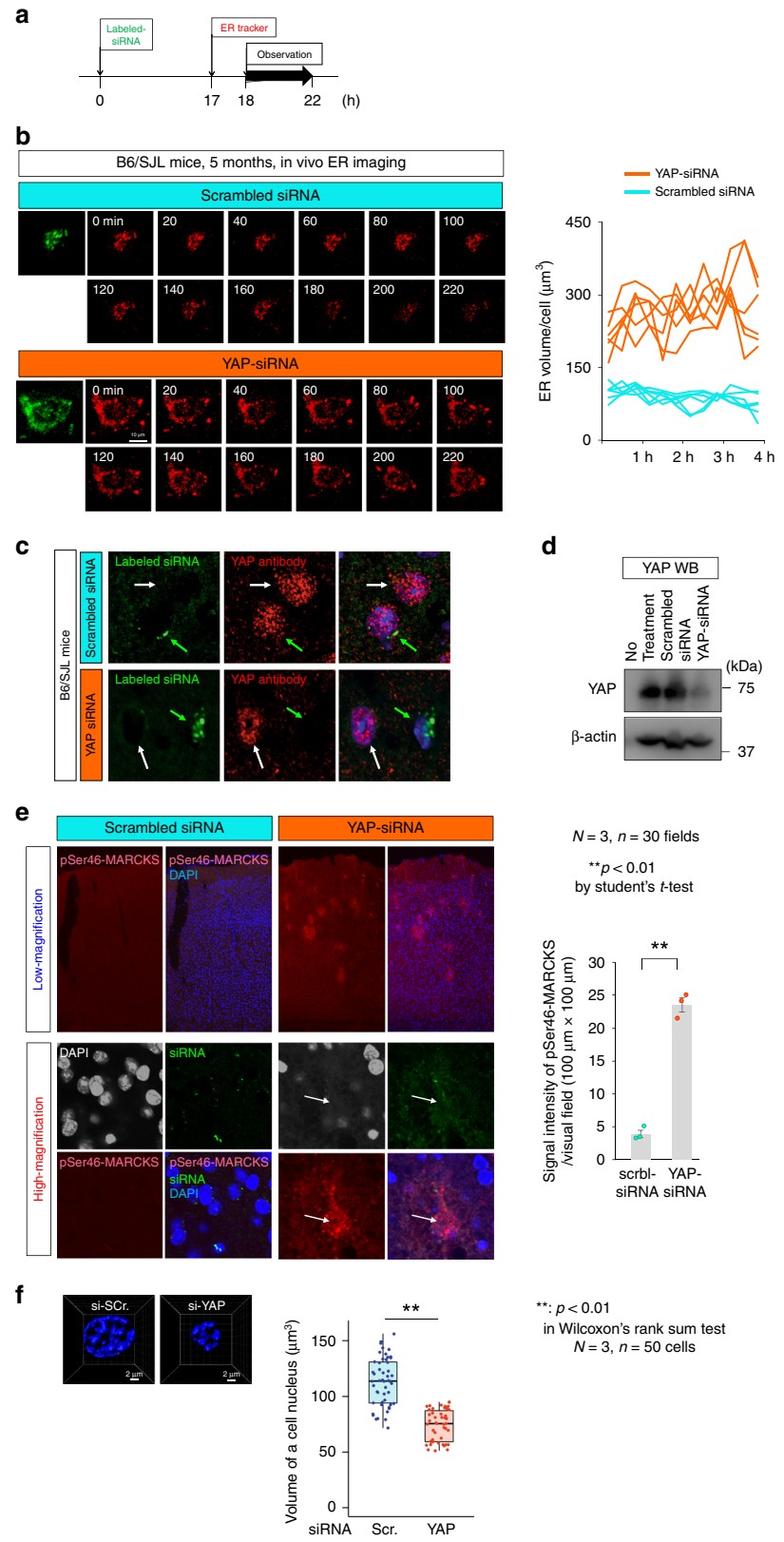

score) was calculated by dividing the number of entries into a new arm different from the previous one by the total number of transfers from one arm to another arm.

**Re-evaluation of anti-pSer46-MARCKS antibody**. *Purification protocol, ELISA, and Immunohistochemistry:* [ENGHVKVNGDA(pS)PA] and [ENGHVKVNG

DASPA] peptides were synthesized, added cysteine at N terminus, and conjugated with KLH (keyhole-limpet hemocyanin). Two rabbits were immunized eight times during nine weeks by the phospho-peptide, and the serum collected at one week after the final immunization was loaded onto non-phospho-peptide column made by [ENGHVKVNGDASPA] peptide and then onto phospho-peptide column made by [ENGHVKVNGDA(pS)PA] peptide. Antibodies bound to each column were eluted by 0.1 M glycine-HCl buffer (pH 2.5).

**Fig. 7 YAP knockdown induces ER ballooning in normal mice. a** Protocol of YAP knockdown in vivo. Labeled YAP-siRNA or scrambled control RNA was transfected into normal mice (B6/SJL), and ER images of siRNA-positive neurons were obtained by two-photon microscopy from 18 h later for 4 h. **b** In vivo time-lapse imaging of normal mice after siRNA transfection by two-photon microscopy. Right graph shows quantitative analysis of ER volume of siRNA-positive neurons. **c** Knockdown of YAP was confirmed by immunohitochemistry after in vivo imaging of two-photon microscopy. Green arrow indicates siRNA-transfected cell, and white arrow indicates non-transfected cell. **d** Knockdown of YAP was confirmed by western blot of mouse cortex tissues dissected after observation by two-photon microscopy. **e** Immunohistochemistry of pSer46-MARCKS after siRNA transfection. Patty immunostains were observed by transfection of YAP-siRNA but not scrambled control siRNA (low magnification). High magnification revealed DAPI-negative cell death surrounded by pSer46-MARCKS signals. Quantitative analysis confirmed the increase of pSer46-MARCKS immunostain signals by YAP-siRNA (right graph). The bar graph indicates average and mean ± S.E.M., together with the corresponding data points. $P$-values were determined by Student's $t$-test, **$p < 0.01$ ($N = 3$ mice, $n = 30$ fields). **f** Quantitative analysis of nuclear volume after YAP-knockdown supported TRIAD necrosis. Left panels: representative nuclei in 3D imaging, right graph: quantitative analysis of a cell nucleus. The box plot shows the median, quartiles and whiskers that represent 1.5× the interquartile range. $P$-values were determined by Wilcoxon's rank sum test, **$p < 0.01$ ($N = 3$ mice, $n = 50$ cells). Source data are provided as a "Source Data file".

ELISA was employed to examine the specific reactivity of anti-pSer46-MARCKS antibody to pSer46-MARCKS as follows. In 50 μL of 20 μg/mL phosphorylated peptide [ENGHVKVNGDA(pS)PA] and non-phosphorylated peptides [ENGHV KVNGDASPA] in PBS were added to each well of the plates (Elisa Plate strips, #FEP-100-008-1, Guangzhou JET Bio-Filtration Products, Co., Ltd, Guangzhou, China) and left for 2 h at room temperature. After washing three times with PBS, 200 μL of 2% BSA in PBS was added to each well and incubated for 2 h at room temperature. After washing the plate twice with PBS, 100 μL of diluted anti-pSer46-MARCKS antibody in PBS was added to each well of the plates and incubated 2 h at room temperature. After washing the wells four times with PBS, 100 μL of HRP-conjugated secondary antibody (ab150077, Abcam, Cambridge, UK) diluted in PBS was added to each well of the plates and incubated 1 h at room temperature. After washing the plate four times with PBS, 100 μL of TMB (EL-TMB Chromogenic Reagent kit, C520026, Sangon Biotech, Shanghai, China) was added to start the reaction and incubated after sufficient color development, and the reaction was stopped with 100 μL of 2 M $H_2SO_4$. Absorbance was measured using a plate reader (iMark, 681130J1, Bio-Rad Laboratories, Hercules, CA, USA) at 450 nm.

Immunohitochemistry were performed as described in the following. Brain tissue sections of 6-month old 5xFAD mice were incubated with the antibodies against Ser46-phosphorylated- or non-phosphorylated-MARCKS at a dilution of 1:2000 overnight at room temperature.

**Immunohistochemistry**. For immunohistochemistry, mouse or human brains were fixed with 4% paraformaldehyde and embedded in paraffin. Sagittal or coronal sections (5 μm thickness) were obtained using a microtome (REM-710, Yamato Kohki Industrial Co., Ltd., Saitama, Japan). Immunohistochemistry was performed using the following primary antibodies: rabbit anti-pSer46-MARCKS (1:2000, ordered from GL Biochem Ltd., Shanghai, China); mouse anti-amyloid β (1:5000, clone 82E1, #10323, IBL, Gunma, Japan); rabbit anti-calnexin (1:200, ab58503, Abcam, Cambridge, UK); mouse anti-KDEL (1:100, ADI-SPA-827, Enzo Life Sciences, NY, USA); rabbit anti-MAP2 (1:200, ab32454, Abcam, Cambridge, UK); rabbit anti-pSer909-LATS1 (1:100, #9157, Cell Signaling Technology, Danvers, MA, USA); rabbit anti-pThr210-PLK1 (1:5000, ab155095, Abcam, Cambridge, UK); rabbit anti-YAP (1:20, sc-15407, Santa Cruz Biotechnology, Dallas, TX, USA); mouse anti-RIP1 (1:200, #610459, BD bioscience, CA, USA); rabbit anti-RIP3 (1:250, ab56164, Abcam, Cambridge, UK); rabbit anti-pSer166-RIP1 (1:400, #44590, Cell Signaling Technology, Danvers, MA, USA); rabbit anti-pSer232-RIP3 (1:100, ab195117, Abcam, Cambridge, UK); rabbit anti-pSer345-MLKL (1:2000, ab196436, Abcam, Cambridge, UK);.Secondary antibodies were as follows: donkey anti-mouse IgG Alexa488 (1:1000, A-21202, Molecular Probes, Eugene, OR, USA), and donkey anti-rabbit IgG Alexa568 (1:1000, A-10042, Molecular Probes, Eugene, OR, USA). Nuclei were stained with DAPI (0.2 μg/mL in PBS, D523, Dojindo Laboratories, Kumamoto, Japan). For multi-labeling, antibodies were labeled by Zenon Secondary Detection-Based Antibody Labeling Kits as follows: anti-cal-nexin, anti-RIP3, anti-pSer166-RIP1 and anti-pSer232-RIP3 (Zenon™ Alexa Fluor™ 555 Rabbit IgG Labeling Kit, Z-25305, Thermo Fisher Scientific, Waltham, MA, USA); anti-MAP2 (Zenon™ Alexa Fluor™ 647 Rabbit IgG Labeling Kit, Z-25308 and Zenon™ Alexa Fluor™ 488 Rabbit IgG Labeling Kit, Z-25302, Thermo Fisher Scientific, Waltham, MA, USA); anti-pSer345-MLKL (Zenon™ Alexa Fluor™ 647 Rabbit IgG Labeling Kit, Z25308 Thermo Fisher Scientific, Waltham, MA, USA); anti-amyloid β (Zenon™ Alexa Fluor™ 488 Mouse IgG₁ Labeling Kit, Z-25002, Thermo Fisher Scientific, Waltham, MA, USA). All images were acquired by fluorescence microscopy (Olympus IX70, Olynpus, Tokyo, Japan) or confocal microscopy (FV1200IXGP44, Olympus, Tokyo, Japan).

**ELISA evaluation of Aβ levels in mouse brains**. We performed sandwich ELISA using Human βAmyloid (1–42) ELISA Kit or Human βAmyloid (1–40) ELISA Kit (298-62401 and 292-62301, FUJIFILM Wako PureChemical Corp., Osaka, Japan). Total proteins extracted from 20 mg of mouse cortex tissues by 1 mL of RIPA buffer (10 mM Tris-HCl pH7.5, 150 mM NaCl, 1 mM EDTA, 1% Triton X-100,

0.1% SDS, 0.1% sodium deoxycholate) were ultra-centrifuged at 100,000 × $g$ at 4 °C for 1 hour. The supernatants were diluted to 100 μL and applied to Aβ1–42 (extract from 100 μg cortex/well) and Aβ1–40 (extract from 500 μg cortex/well) ELISA plates and measured following the manufacturer's instructions.

The ELISA plates were incubated at 4 °C overnight, washed five times with 1x wash solution, added with 100 μL of HRP-conjugated antibody solution, and incubated at 4 °C for 1 or 2 h in Aβ1–42 or Aβ1–40 ELISA kits, respectively. After washed five times by 1x wash solution, the ELISA plates were added with 100 μL of TMB solution, incubated at room temperature for 30 min, and added with 100 μL of stop solution. Absorbance was measured at 450 nm using a plate reader (SPARK 10 M, TECAN, Grodig, Austria).

**Immuno-electron microscopy**. The tissues were fixed with 4% paraformaldehyde for 12 h, followed by the cryo-protective treatment with 30% sucrose. The frozen tissue blocks in the cryo-compound were sliced with 20 μm thickness with cryostat. Sections were incubated with the 5% Block Ace (UKB80, DS Pharma Biomedical, Osaka, Japan) solution with 0.01% Saponin in 0.1 M PB for an hour, and stained with primary rabbit anti-pSer46-MARCKS (1:1000, ordered from GL Biochem Ltd., Shanghai, China) for 72 h at 4 °C, followed by the incubation with nanogold conjugated goat anti-rabbit secondary antibody (1:100, N-24916, Thermo Fisher Scientific, Waltham, MA, USA) for 24 h at 4 °C. After 2.5% glutaraldehyde fixation in PB, nanogold signals were enhanced with R-Gent SE-EM Silver Enhancement Reagents (500.033, Aurion, Wageningen, Netherlands) for 40 min at 25 °C. Stained sections were post-fixed with 1.0% OsO4 for 90 min at 25 °C, dehydrated through graded series of ethanol and embedded in Epon. Ultrathin sections (70 nm) were prepared with ultramicrotome (UC7, Leica, Wetzlar, Germany) and stained with uranyl acetate and lead citrate. The sections were observed under a transmission EM (JEOL model 1400 plus, JEOL Ltd., Tokyo, Japan).

**Immunoprecipitation**. Mouse cerebral cortex was lysed in a homogenizer with RIPA buffer (10 mM Tris–HCl, pH 7.5, 150 mM NaCl, 1 mM EDTA, 1% Triton X-100, 0.1% SDS, 0.1% DOC, 0.5% protease inhibitor cocktail (539134, Calbiochem, San Diego, CA, USA)). Lysates were rotated for 60 min at 4 °C, and then centrifuged (12,000 $g$ × 1 min at 4 °C). Supernatant (250 μg) was incubated with a 50% slurry of protein G–Sepharose beads (100 μL, 17061801, GE Healthcare,, Chicago, IL, USA), followed by centrifugation (2000 $g$ × 3 min at 4 °C). Supernatants were incubated with 1 μg of antibody for 16 h at 4 °C with rotation. Antibodies were as follows: rabbit anti-YAP (1:100, #14074, Cell Signaling Technology, Danvers, MA, USA); mouse anti-amyloid β (1:5000, clone 82E1, #10323, IBL, Gunma, Japan). Protein G–Sepharose was added to samples and rotated for 4 h at 4 °C, and then the beads were washed three times with RIPA buffer. Equal volume of sample buffer (125 mM Tris–HCl, pH 6.8, 4% SDS, 10% glycerol, 0.005% BPB, 5% 2-mercaptoethanol) was added, and the samples were boiled at 100 °C for 10 min before SDS–PAGE.

**Dot blot**. Protein concentrations in samples were measured using the BCA Protein Assay Regent (Micro BCA Protein Assay Reagent kit, 23225, Thermo Fisher Scientific, Waltham, MA, USA). After the membranes (Immobilon-P, IPVH00010, Merck Millipore, Burlington, MA, USA) were washed by TBS (20 mM Tris-HCl/ pH 7.5, 500 mM NaCl), samples of 2.5 μg/25 μL were dropped on membranes using Bio-Dot Apparatus (1706545, Bio-Rad Laboratories Laboratories, CA, USA) and left to stand overnight. Next, the membranes were blocked with 5% skim milk in TBST (10 mM Tris/HCl pH 8.0, 150 mM NaCl, 0.05% Tween-20), and reacted with the following primary and secondary antibodies diluted in Can Get Signal solution (NKB-101, Toyobo, Osaka, Japan). ECL prime (RPN2232, GE Healthcare, Chicago, IL, USA) or ECL select (RPN2235, GE Healthcare, Chicago, IL, USA) were used to detect the bands using LAS500 (29005063, GE Healthcare, Chicago, IL, USA). Primary and secondary antibodies were diluted as follows: mouse anti-amyloid β, (1:5000, clone 82E1, #10323, IBL, Gunma, Japan); HRP-conjugated anti-mouse IgG (1:5000, NA931VA, GE Healthcare, Chicago, IL, USA).

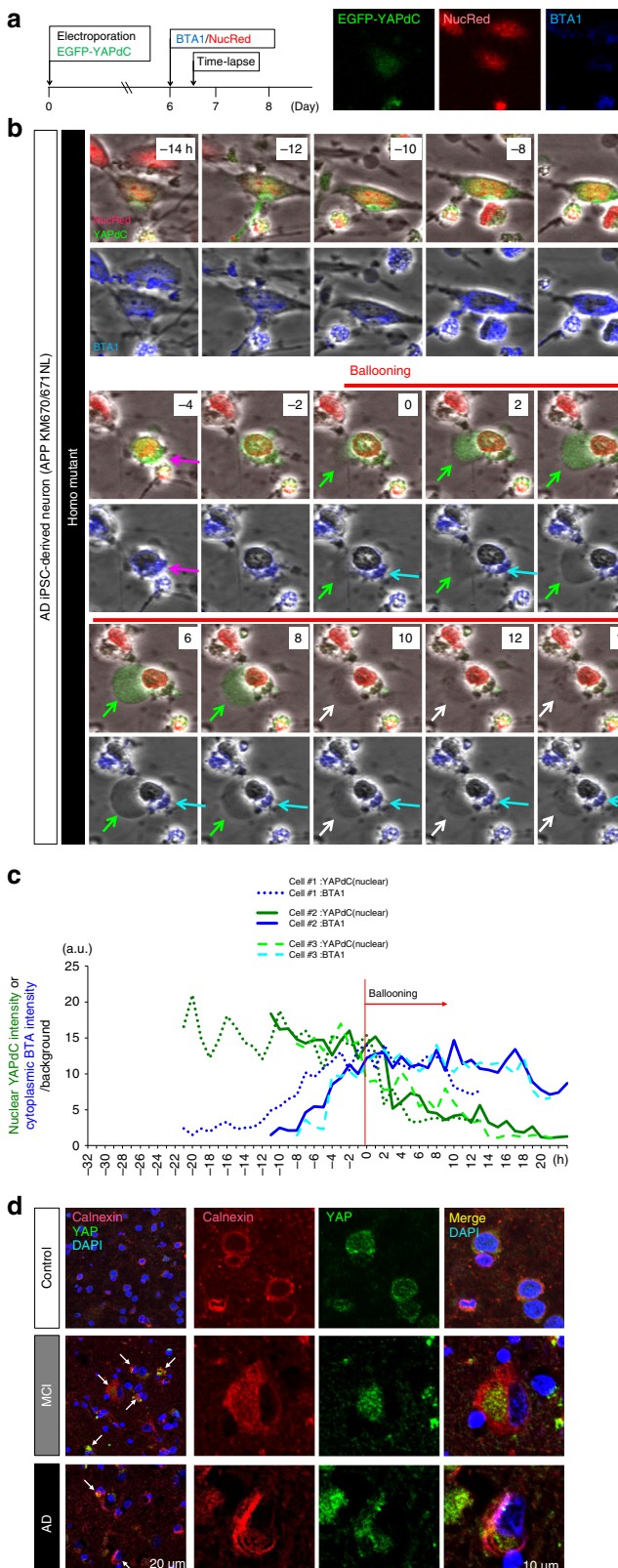

**Fig. 8 Time lapse imaging of a single neuron elucidates pathological cascade. a** Time lapse imaging protocol of a single neuron to analyze relationship between intracellular localization of YAP, intracellular Aβ and ER ballooning. Representative images of the same cells (similar to the cells in **b**) are shown in right panels. **b** Time lapse imaging of an iPSC-derived neurons carrying APP mutations (homozygous mutants carrying APP KM670/671NL). Nuclear YAP was shifted to intracellular Aβ in the cytoplasm (magenta arrow) and further to ballooned ER (green arrow). YAP was released to extracellular space via leakage of ballooned ER (white arrow) while intracellular Aβ remains as aggregates (blue arrow). The details were described in the text. **c** Chronological change of nuclear YAPdeltaC intensity and cytoplasmic BTA intensity in three iPSC-derived neurons carrying APP mutations. **d** Immunohistochemistry of human cerebral cortex with anti-calnexin, an ER membrane marker, and anti-YAP antibodies. Abnormal localizations of YAP in the cytoplasm or ballooned ER were observed frequently observed in MCI patients and at a low frequency in AD patients (white arrow), consistently with the findings in iPSC-derived neurons carrying APP mutations. Source data are provided as a "Source Data file".

following primary and secondary antibodies diluted in Can Get Signal solution (NKB-101, Toyobo, Osaka, Japan). Bands were visualized using ECL prime (RPN2232, GE Healthcare, Chicago, IL, USA) or ECL select (RPN2235, GE Healthcare, Chicago, IL, USA). Primary and secondary antibodies were diluted as follows: rabbit-anti-YAP (H-125) (1:3000, sc-15407, Santa Cruz Biotechnology, Dallas, TX, USA); β-actin(C-4) (1:3000, sc-47778, Santa Cruz Biotechnology, Dallas, TX, USA); mouse anti-amyloid β, (1:1000, clone 82E1, #10323, IBL, Gunma, Japan); rabbit-anti-YAPdeltaC (1:9000)[17]; mouse anti-RIP1 (1:1000, 610459, BD bioscience, CA, USA); rabbit anti-RIP3 (1:1000, ab56164, Abcam, Cambridge, UK); anti-pSer166-RIP1 (1:1000, #44590, Cell Signaling Technology, Danvers, MA, USA); anti-pSer232-RIP3 (1:1000, ab195117, abcam, Cambridge, UK); anti-pSer46-MARCKS antibody (1:100000, ordered from GL Biochem Ltd., Shanghai, China); anti-histone H3 antibody (1:1000, 630767, Merck, Darmstadt, Germany) HRP-conjugated anti-mouse IgG (1:3000, NA931VA, GE Healthcare, Chicago, IL, USA); and HRP-conjugated anti-rabbit IgG (1:3000, NA934VS, GE Healthcare, Chicago, IL, USA).

**Immunocytochemistry**. iPS-derived neurons were fixed in 4% PFA, and then permeabilized by incubation with 0.1% Triton X-100 in PBS for 10 min at room temperature (RT). After blocking with blocking buffer (50 mM Tris-HCl pH 6.8, 150 mM NaCl, and 0.1% Triton X-100) containing 5 mg/mL BSA for 60 min at RT, sections were incubated with primary antibody for 60 min or 180 min (only for 6E10), and finally with secondary antibodies for 60 min at RT. The antibodies used for immunocytochemistry were diluted as follows: rabbit-anti-YAP (1:100, #14074 S, Cell Signaling Technology, Danvers, MA, USA), which was raised against amino acids around Pro435 of human YAP isoform 1; rabbit-anti-YAP (1:200, sc-15407, Santa Cruz Biotechnology, Dallas, TX, USA), which was raised against amino acids 206–330 of human YAP; rabbit-anti-YAPdeltaC (1:2000)[17]; mouse-anti-Aβ (1:250, clone 6E10, SIG-39300, Covance, NJ, USA); anti-pSer46-MARCKS antibody (1:2000, ordered from GL Biochem Ltd., Shanghai, China); Cy3-conjugated anti-mouse IgG (1:500, 715-165-150, Jackson Laboratory, Bar Harbor, ME, USA); and Alexa Fluor 488–conjugated anti-rabbit IgG (1:1000, A11008, Molecular Probes, Eugene, OR, USA).

**TEAD-YAP transcriptional activity**. Neurospheres differentiated from human iPS cells (with or without APP mutations)[37] were dissociated in TrypLE™ Select (12563-011, Thermo Fisher Scientific, Waltham, MA, USA) containing 10 μM Y27632 (253-00513, Wako, Osaka, Japan). In total 4 × 10⁵ cells, centrifuged, and suspended in 20 μL nucleofector solution (P3 Primary Cell 4D-Nucleofector™ X Kit, V4XP-3012, LONZA, NJ, USA). In total 1 μg of pLL3.7-ires-GFP-TEAD-respon-sive-H2B-mCherry plasmid[38] (generous gift from Prof. Yutaka Hata, Tokyo Medical and Dental University) was added to the cell suspension and electroporated into cells by 4D-Nucleofector (pulse program: CV-110) (4D-Nucleofector Core Unit, #AAF-1002B, LONZA, NJ, USA). The electroporated cells were cultured on Lab-Tek II chambered coverglass coated with poly-L-ornithine (P3655, Sigma-Aldrich, St. Louis, MO, USA) and laminin (23016015, Thermo Fisher Scientific, Waltham, MA, USA) in DMEM/F12 (D6421, Sigma-Aldrich, St. Louis, MO, USA) supplemented with B27 (17504044, Thermo Fisher Scientific, Waltham, MA, USA), Glutamax (35050061, Thermo Fisher Scientific, Waltham, MA, USA), and penicillin/streptomycin (15140-122, Thermo Fisher Scientific, Waltham, MA, USA). After differentiation to neurons for seven days, time-lapse images of pTEAD-driven mCherry and pCMV-driven EGFP were acquired by Olympus FV10i-W (Olympus, Tokyo, Japan) at 30 min intervals for 36 h. Cells were co-

**Western blot**. Protein concentration of samples was measured using the BCA Protein Assay Regent (Micro BCA Protein Assay Reagent kit, 23225, Thermo Fisher Scientific, Waltham, MA, USA). After samples were separated by SDS-PAGE, they were transferred onto polyvinylidene difluoride membranes (Immobilon-P, IPVH00010, Merck Millipore, Burlington, MA, USA) using the semi-dry method. Next, the membranes were blocked with 5% skim milk in TBST (10 mM Tris/HCl pH 8.0, 150 mM NaCl, 0.05% Tween-20), and reacted with the

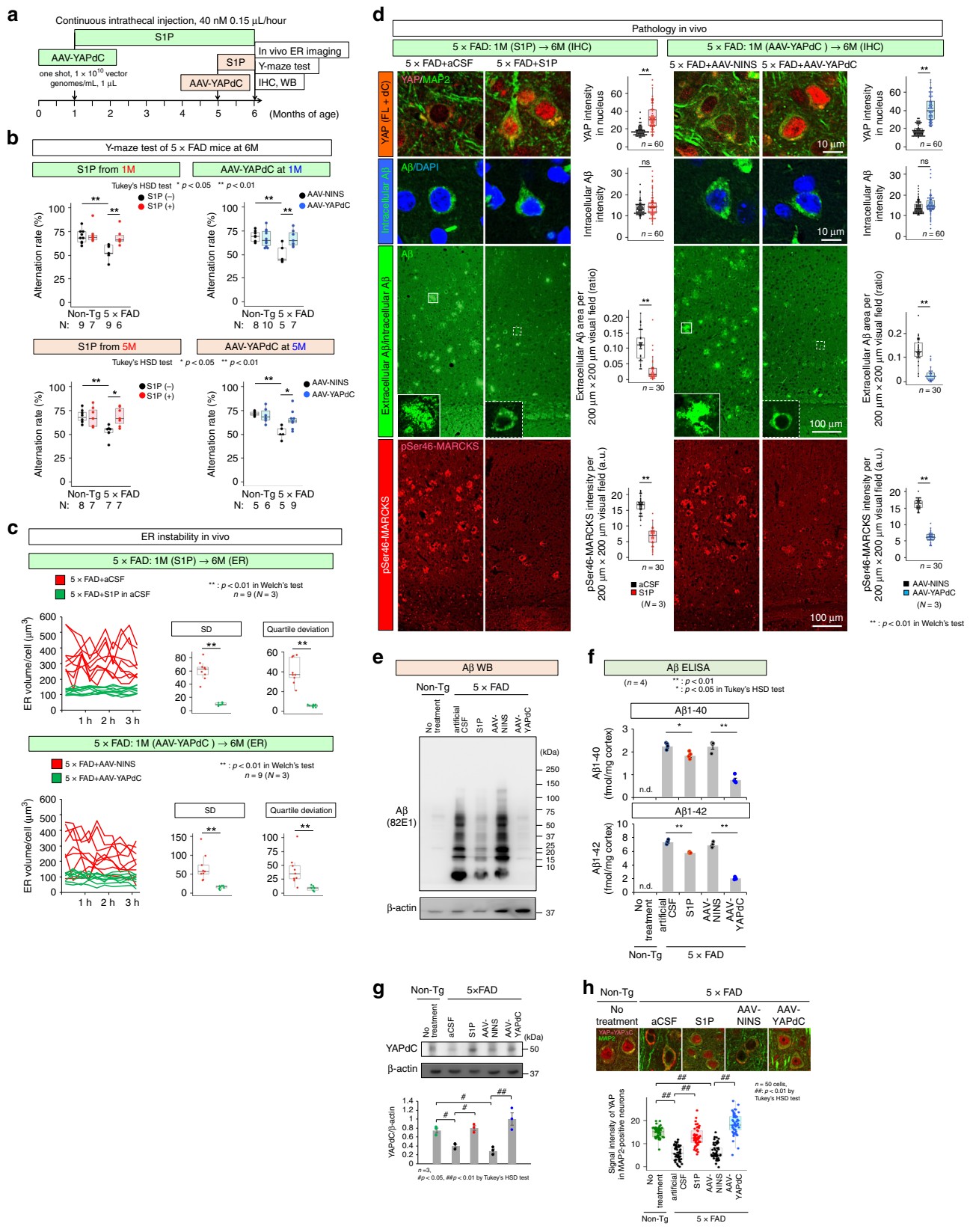

stained with ER-Tracker™ Blue-White DPX (#E12537, Thermo Fisher Scientific, MA, USA) and ER signals were obtained in parallel. Measurement of signal intensity was performed using Fluoview (Olympus, Tokyo, Japan) software. The chamber was kept at 37 °C with 5% $CO_2$ during the experiment.

**siRNA-mediated knockdown of YAP in iPS-derived neurons in vitro.** In total 5 pmol (67.6 ng) of human YAP-siRNA (sc-38637, Santa Cruz Biotechnology, Dallas, TX, USA) or 5 pmol of Trilencer-27 universal scrambled negative control siRNA duplex (#SR30004, OriGene, Rockville, MD, USA) was transfected into human

**Fig. 9 S1P and YAPdeltaC rescue ER instability and cognitive impairment in AD model mice. a** Experimental protocol for the rescue effect of S1P or YAPdeltaC (YAPdC) on ER instability and cell death in 5xFAD mice. Two protocols, administration before symptomatic onset (1 month) or administration just after onset (5 months), were used. AAV-NINS: AAV-CMV-no insert. **b** Alteration rate in Y-maze test of 6-month-old 5xFAD mice that had been treated with S1P from 1 month (upper left panel) or 5 months (lower left panel) or injected with AAV-YAPdeltaC at 1 month (upper right panel) or 5 months (lower right panel). *P*-values were determined by Tukey's HSD test, *$p < 0.05$, **$p < 0.01$. N: shown below graphs. **c** ER instability was rescued at 6 months following treatment from 1 month with S1P or YAPdeltaC. *P*-values were determined by Welch's test, **$p < 0.01$ ($N = 3$ mice, $n = 9$ cells). **d** Pathological examination at 6 months following treatment from 1 month with S1P or YAPdeltaC. Staining of YAP with the sc-15407 antibody detecting YAP-FL and YAPdeltaC, intracellular/extracellular Aβ, and pSer46MARCKS is shown. Right graphs show quantification of the four stains before and after the treatment. P-values were determined by Welch's test, **$p < 0.01$ ($N = 3$ mice, $n = 30$ or $60$). **e** Western blot with anti-Aβ antibody (82E1) confirmed the effect of S1P and AAV-YAPdC on Aβ burden in 5xFAD mice. **f** ELISA of Aβ1-40 and Aβ1–42 consistently showed the effect of S1P and AAV-YAPdC reducing Aβ burden in 5xFAD mice. *P*-values were determined by Tukey's HSD test, *$p < 0.05$, **$p < 0.01$ ($n = 4$ mice). **g** Western blot confirming the increase of YAPdeltaC protein in cerebral cortex tissue of 5xFAD mice after AAV-YAPdeltaC infection. S1P also increased YAPdeltaC. *P*-values were determined by Tukey's HSD test, #$p < 0.05$, ##$p < 0.01$ ($n = 3$ tests). **h** Immunohistochemistry of cerebral cortex tissue of 5xFAD mice supported the increase of total YAP after S1P and AAV-YAPdeltaC treatments. *P*-values were determined by Tukey's HSD test, ##$p < 0.01$ ($n = 50$ cells). Box plots show the median, quartiles and whiskers that represent 1.5× the interquartile range. Bar graphs indicate average and mean S.E.M., together with the corresponding data points. Source data are provided as a "Source Data file".

neurons differentiated from normal iPS cells (ASE-9203, Applied StemCell, Milpitas, CA, USA) using Viromer® BLUE (TT100300, OriGene, Rockville, MD, USA). Before transfection, siRNA was labeled with Label IT® siRNA Tracker™ Fluorescein Kit without Transfection Reagent (MIR7216, Mirus, WI, USA) according to the manufacturer's procedures. Eighteen hours later, cells were stained with ER-Tracker™ Red (BODIPY™ TR Glibenclamide) (E34250, Thermo Fisher Scientific, MA, USA) and NucRed™ Live 647 ReadyProbes™ Reagent (R37106, Thermo Fisher Scientific, MA, USA) for 60 min at 37 °C. Time-lapse images of iPS-derived neurons were acquired at ×60 magnification on an Olympus FV10i-W (Olympus, Tokyo, Japan) at 15 min intervals for 24 h. The chamber was kept at 37 °C with 5% $CO_2$. The ratio of cell death patterns was counted cells transfected labeled-siRNA. To validate knockdown efficiency of siRNA, cells were fixed and collected. Each sample was used for Immunocytochemistry and western blot.

**siRNA-mediated knockdown of YAP in mice in vivo.** Before siRNA injection, siRNA was labeled with Label IT® siRNA Tracker™ Fluorescein Kit without Transfection Reagent (MIR7216, Mirus, WI, USA) according to the manufactural procedures. Under anesthesia with 1% isoflurane, we injected 300 ng Fluorescein-labeled-siRNA (mouse YAP-siRNA, sc-38638, Santa Cruz Biotechnology, Dallas, TX, USA) or Trilencer-27 universal scrambled negative control siRNA duplex (sc-38637, Santa Cruz Biotechnology, Dallas, TX, USA) into retrosplenial cortex (anteroposterior, −2.0 mm form bregma; lateral, 0.6 mm; depth, 1 mm) at a volume of 1 μl using in vivo jetPEI (201-10 G, Polyplus-transfection, Illkirch, France) at 5 months of age (21 weeks). After 16 h, live-cell imaging of ER was performed by two-photon microscopy followed by the method listed above ("In vivo imaging of neuronal ER and Aβ").

After dissection, mouse cerebral cortexes were fixed in 4% PFA for overnight at 4 °C. Tissue sections were prepared at 30 μm thickness using microtome (MICROM HM650V, Thermo Fisher Scientific, Waltham, MA, USA), and stained in floating condition. In brief, sections were incubated with blocking solution (10% FBS and 0.3%triton-X in PBS) for 30 min at RT, and with a primary antibody (anti-YAP antibody (1:100, sc-15407, Santa Cruz Biotechnology, Dallas, TX, USA); pSer46-MARCKS (1:2000, ordered from GL Biochem Ltd., Shanghai, China)) overnight at 4 °C. After three times of washing with PBS for 5 min, the sections were incubated with goat-anti rabbit IgG Alexa568 for 1 h at room temperature, and with DAPI for 5 min.

**Pathological cascade analysis of a single cell neuron.** pEGFP-YAPdeltaC was generated by subcloning EcoRI-SalI fragment digested from pBS-YAPdeltaC[17] into pEGFP-C1 (#6084-1, Clontech, Mountain View, CA, USA). Neurospheres differentiated from human iPS cells (with or without APP mutations)[37] were dissociated in TrypLE™ Select (12563-011, Thermo Fisher Scientific, Waltham, MA, USA) containing 10 μM Y27632. $4 \times 10^5$ cells, centrifuged, and resuspended in 20 μL nucleofector solution (P3 Primary Cell 4D-Nucleofector™ X Kit, V4XP-3012, LONZA, NJ, USA). One μg of pEGFP-YAPdeltaC was added to the cell suspension and electroporated into cells by 4D-Nucleofector (pulse program: CV-110) (4D-Nucleofector Core Unit, AAF-1002B, LONZA, fNJ, USA). The electroporated cells were cultured on Lab-Tek II chambered coverglass coated with poly-L-ornithine (P3655, Sigma-Aldrich, St. Louis, MO, USA) and laminin (23016015, Thermo Fisher Scientific, Waltham, MA, USA) in DMEM/F12 (D6421, Sigma-Aldrich, St. Louis, MO, USA) supplemented with B27 (17504044, Thermo Fisher Scientific, Waltham, MA, USA), Glutamax (35050061, Thermo Fisher Scientific, Waltham, MA, USA), and penicillin/streptomycin (15140-122, Thermo Fisher Scientific, Waltham, MA, USA). After differentiation to neurons for 6 days, time-lapse images of EGFP-YAPdeltaC were acquired by Olympus FV10i-W (Olympus, Tokyo, Japan) at 30 min intervals for 36 h. Cells were co-stained with NucRed™ Live 647 ReadyProbes™ Reagent (R37106, Thermo Fisher Scientific, MA, USA) and 100 nM

2-(4′-methylaminophenyl) benzothiazole (BTA1, B9934, Sigma-Aldrich, MO, USA). The chamber was kept at 37 °C with 5% $CO_2$ during the experiment. Measurement of signal intensity was performed using the ImageJ software (Ver. 1.45 s).

**Generation of AAV-YAPdeltaC.** The AAV vector plasmid carries an expression cassette consisting of a human CMV promoter, the first intron of human growth hormone 1, cDNA encoding rat YAPdeltaC (ins61; accession no. DQ186898.2), woodchuck hepatitis virus post-transcriptional regulatory element (WPRE), and a simian virus 40 polyadenylation signal sequence (SV40 poly[A]) between the inverted terminal repeats of the AAV1 genome. To generate the vectors, AAV2 rep and AAV1 vp expression plasmids, and an adenoviral helper plasmid (pHelper, 240071-54, Agilent Technologies, CA, USA) were co-transfected into HEK293 cells by the calcium phosphate co-precipitation method. The recombinant viruses were purified by isolation from two sequential continuous CsCl gradients. Viral titers were determined by qPCR.

**Intrathecal injection of AAV and S1P.** In pharmacological rescue experiments, sphingosine-1-phosphate (S1P, 40 nM Sigma-Aldrich, St. Louis, MO, USA) or artificial cerebrospinal fluid (3525, aCSF, R&D systems, MN, USA) was administered to mice at 4 weeks old into the subarachnoid space via osmotic pump (0.15 μL/h, model 2006, ALZET, Cupertino, Canada) for 28 days. For the YAP rescue experiment, AAV1-YAPdeltaC or AAV-CMV-NINS (titer: $1 \times 10^{10}$ vector genomes/mL, 1 μL) was injected into retrosplenialdysgranular cortex at −2.0 mm from bregma, mediolateral 0.6 mm, depth 1 mm. Two days after drug injection, imaging was performed 10 times at 20 min intervals using the following method.

**In vivo imaging of neuronal ER and Aβ.** The skull was thinned with a high-speed micro-drill at the surface of the mouse splenial cortex[57]. The head of each mouse was immobilized by attaching the head plate to a custom machine stage mounted on the microscope table. Two-photon imaging was performed using FV1000MPE2 (Olympus, Tokyo, Japan) equipped with an upright microscope (BX61WI, Olympus, Tokyo, Japan), a water-immersion objective lens (XLPlanN25xW; numerical aperture, 1.05, Olympus, Tokyo, Japan), and a pulsed laser (MaiTaiHP DeepSee, Spectra Physics, Santa Clara, CA, USA). Four hours before imaging, BTA1 (100 nM, B-9934, Sigma-Aldrich, MO, USA) and ER-Tracker™ Red (100 nM, E34250, Thermo Fisher Scientific, MA, USA) were injected in a volume of 1 μL into RSD at −2.0 mm from bregma, mediolateral ±0.6 mm, depth 1 mm, under anesthesia with 1% isoflurane. Both BTA1 and ER were excited at 750 nm and scanned at 495–540 nm and 575–630 nm, respectively. High-magnification imaging (101.28 × 101.28 μm; 1024 × 1024 pixels; 1 μm Z-step; 60–80 slices along Z-axis) of the cortical layer I in RSD was performed with a 2 × digital zoom through the thinned-skull window in the retrosplenial cortex[57]. Images of BTA1 and ER-Tracker™ Red were analyzed according to three parameters: ER and BTA1 signal intensity, ER or BTA1 puncta volume, and number of ER-positive cells or BTA1-positive cells per imaging volume. Measurement of ER signal intensity was performed using the ImageJ software (Ver. 1.45 s). Total ER puncta volumes belonging to a single neuron were quantified by IMARIS (Bitplane, Zurich, Switzerland).

**Neurons derived from genome-edited human iPS cells.** Human normal iPS cells (ASE-9203, Applied StemCell, Milpitas, CA, USA) were transfected with a mixture of plasmids expressing gRNA (5′-GGAGATCTCTGAAGTGAAGATGG-3′) and the Cas9 gene along with single-stranded oligodeoxynucleotides (for human APP KM670/671NL, 5′-TTGGTTGTCCTGCATACTTTAATTATGATGTAATACAG GTTCTGGGGTTGACAAATATCAAGACGGAGGAGATCTCTGAAGTGAATCT GGATGCAGAATTCCGACATGACTCAGGATATGAAGTTCATCATCAAAAA

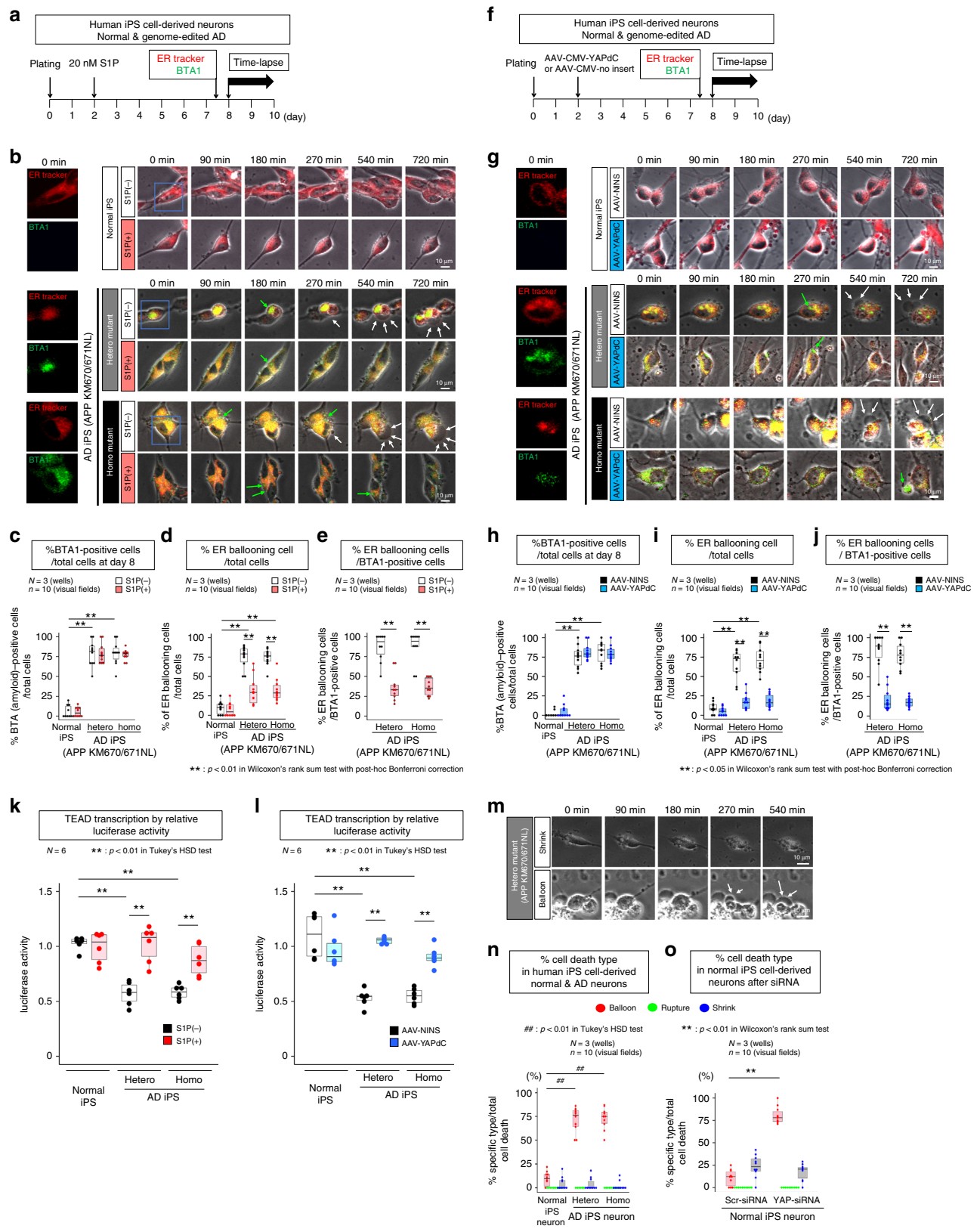

TTGGTACGTAAAATAATTTACCTCTTTC-3′) for donor DNA. The Cas9 gene was fused to the 2 A peptide and GFP gene. Cells were electroporated using a Neon system (MPK5000, Thermo Fisher Scientific Inc., MA, USA) with the following conditions: 1200 V, 30 ms, one pulse. Cells were selected with 0.4 µg/mL puromycin for 24–48 h after transfection, and then subjected to a colony cloning process by picking and seeding each visible GFP-positive colony into a well of a 96-well plate. The cells were allowed to grow for 7–10 days, or until a conveniently

sized colony was formed. A portion of cells from each colony was subjected to genotype analysis. Briefly, genomic DNA from single-cell colonies was isolated and used to amplify a 308 bp DNA fragment using primers 5′-GCATGTATTTAAAG GCAGCAGAAGC-3′ and 5′-CAATGCTTGCCTATAGGATTACCATGAAAAC ATG-3′. PCR fragments were subjected to Sanger sequencing. Positive clones were expanded, and a portion of cells was resubmitted for sequencing to confirm the desired genotype. Primers are listed in Supplementary Table 3.

**Fig. 10 S1P and YAPdeltaC rescue ER instability and necrosis in genome-edited iPSC-derived AD neurons. a** Protocol for the rescue effect of S1P on ER instability and cell death. **b** Time-lapse images of non-treated iPSC-derived AD neurons exhibited ER ballooning (white arrows) and rupture. S1P treatment stabilized the ER and decreased necrosis. Aβ accumulation in ER (yellow area) and the leak to cytoplasm (green arrow). Representative images are shown in Supplementary Movie 7-12. **c** S1P did not change %BTA1-positive iPSC-derived AD neurons. **d, e** Suppressive effect of S1P on ER ballooning and necrosis of iPSC-derived AD neurons **d** and BTA1-positive neurons **e**. **f** Protocol for the rescue effect of YAPdeltaC on ER instability and cell death. AAV-NINS: AAV-CMV-no inset. **g** Time-lapse images showed ER ballooning (white arrows) and rupture of iPSC-derived AD neurons, which were suppressed by YAPdeltaC expression. Aβ was mainly accumulated in ER (yellow area), but a portion leaked into the cytoplasm (green arrow). Representative images are shown in Supplementary Movie 13-18. **h** YAPdeltaC did not change %BTA1-positive iPSC-derived AD neurons. **i** Suppressive effect of YAPdeltaC on ER ballooning and necrosis in iPSC-derived AD neurons **i** and BTA1-positive neurons **j**. **k, l**. Rescue of transcriptional function, as determined using TEAD-responsive element reporter plasmid in iPSC-derived neurons. S1P **k** and YAPdeltaC **l** restored TEAD-YAP/YAPdeltaC-dependent transcription, which was suppressed in iPSC-derived AD neurons. **p < 0.01 (N = 6 wells), Tukey's HSD test. **m** Time-lapse images of shrinkage apoptosis and ballooning necrosis (arrow). **n** The ratio of each type of cell death (shrinkage apoptosis, rupture necrosis and ballooning necrosis) to total neurons that occurred naturally during 24 h of time-lapse imaging in normal and iPSC-derived AD neurons. ##p < 0.01 (N = 3 wells, n = 10 visual fields), Tukey's HSD test. **o**. The ratio of each type of cell death to total neurons that occurred during 24 h after transfection of YAP-siRNA or scrambled-siRNA. **p < 0.01 (N = 3 wells, n = 10 visual fields), Wilcoxon's rank sum test. **c, d, e, h, i, j** **p < 0.01 (N = 3 wells, n = 10 visual fields) by Wilcoxon's rank–sum test with post-hoc Bonferroni correction. Box plots show the median, quartiles and whiskers that represent 1.5× the interquartile range. Source data are provided as a "Source Data file".

**Live imaging of iPSC-derived neurons**. Human normal iPS cells and mutant human iPS cells were plated on a 6 cm dish with 3 μM SB431542, 3 μM CHIR99021, and 3 μM dorsomorphin, and cultured for 6 days. Next, iPS cells were dissociated into single cells using TrypLE™ Select (12563-011, Thermo Fisher Scientific, MA, USA) containing 10 μM Y27632 (253-00513, Wako, Osaka, Japan). To form neurospheres, the dissociated cells were cultured in KBM medium (16050100, KHOJIN BIO, Saitama, Japan) with 20 ng/mL Human-FGF-basic (100-18B, Peprotech, London, UK), 10 ng/mL Recombinant Human LIF (NU0013-1, Nacalai, Kyoto, Japan), 10 μM Y27632 (253-00513, Wako, Osaka, Japan), 3 μM CHIR99021 (13122, Cayman Chemical, Ann Arbor, MI, USA), and 2 μM SB431542 (13031, Cayman Chemical, Ann Arbor, MI, USA) under suspension culture conditions in a 10 cm cell-repellent dish. Neurospheres were passaged twice every 7 days, and then dissociated in TrypLE™ Select containing 10 μM Y27632 (253-00513, Wako, Osaka, Japan). Dissociated cells were re-seeded onto coverslips coated with poly-L-ornithine (P3655, Sigma-Aldrich, St. Louis, MO, USA) and laminin (23016015, Thermo Fisher Scientific, Waltham, MA, USA) in 8-well chambers or 6-well plates with DMEM/F12 (D6421, Sigma-Aldrich, St. Louis, MO, USA) supplemented with B27 (17504044, Thermo Fisher Scientific, Waltham, MA, USA), Glutamax (35050061, Thermo Fisher Scientific, Waltham, MA, USA), and penicillin/streptomycin (15140-122, Thermo Fisher Scientific, Waltham, MA, USA). Two days later, cells were infected with AAV-CMV-YAPdeltaC-ins61 or AAV-CMV-NINS (MOI: 5000) or treated with 20 nM S1P (S9666, Sigma-Aldrich, MO, USA). Six days after viral or drug application, cells were stained with BTA1 (B-9934, Sigma-Aldrich, St. Louis, MO, USA) and ER-Tracker™ Red (BODIPY™ TR Glibenclamide) for 60 min at 37 °C, and then subjected to live-cell imaging (E34250, Thermo Fisher Scientific, Waltham, MA, USA). Time-lapse images of iPS-derived neurons were acquired at ×60 magnification on an Olympus FV10i-W (Olympus, Tokyo, Japan) at 30 min intervals for 48 h. The chamber was kept at 37 °C with 5% CO₂. The ratio of cell death patterns was counted 12 h after the start of time-lapse image acquisition.

**Luciferase assay with iPSC-derived neurons**. iPSC-derived neurons (2 × 10⁴ cells) were transfected with 10 μg of 8xGTIIC-luciferase reporter (34615, Addgene, Watertown, MA, USA) and 10 μg of pGL4.74[hRluc/TK] Vector (E6921, Promega, Madison, WI, USA) using Lipofectamine LTX with Plus Reagent (14338100, Thermo Fisher Scientific, Waltham, MA, USA). The 8xGTIIC-luciferase reporter plasmid possesses eight synthetic TEAD-binding sites upstream of the luciferase gene, making it YAP/TAZ-responsive; this construct was generated by adding four more TEAD-binding sites to 4XGTIIC-Lux, originally created by Ian Farrance[58] (https://www.addgene.org/34615/). pGL4.74[hRluc/TK] encodes the luciferase reporter gene hRluc (*Renilla reniformis*). After 48 h transfection, an equal volume of Dual-Glo Luciferase Reagent (E2920, Promega, Madison, WI, USA) was added to each well. Firefly luminescence was measured on a Spark 10 M multimode microplate reader (TECAN, Männedorf, Switzerland) after incubation for 20 min. For *Renilla* luminescence, an equal volume of Dual-Glo Stop & Glo Reagent (E2920, Promega, Madison, WI, USA) was added to each well before Dual-Glo Luciferase Reagent, mixed, and measured on a Spark 10 M. For the recovery experiments, AAV-CMV-YAPdeltaC-ins61 or AAV-CMV-NINS (MOI: 5000) or 20 nM S1P (S9666, Sigma-Aldrich, St. Louis, MO, USA) was added to the culture medium 4 days before plasmid transfection.

**Electron microscopy of ballooning neurons in mice and human**. Mouse and human brain samples were fixed with 2.5% glutaraldehyde in 0.1 M phosphate-buffered saline (PBS) for 2 h, incubated with 1% OsO4 buffered with 0.1 M PBS for 2 h, and dehydrated in a series of graded concentrations of ethanol (50, 70, 80, 90, 100, 100, 100, and 100%), and embedded in Epon812 (E14120, science services,

München, Germany). Semi-thin (1 μm) sections for light microscopy were collected on glass slides and stained for 30 s with toluidine blue. Ultrathin (90 nm) sections were collected on copper grids, double-stained with uranyl acetate and lead citrate. Images were obtained by transmission electron microscopy (H-7100, Hitachi, Hitachinaka, Ibaraki, Japan).

Human samples (cerebral neocortex) was collected at autopsy and directly fixed in 4% paraformaldehyde two overnight. The sliced brain samples were preserved in 20% sucrose contained phosphate buffer in 4 °C until the process for ultrastructural observation. Following procedures were same as mice tissue sample preparation.

**Induction of apoptosis and necrosis of primary cortical neurons**. Mouse primary cortical neurons were prepared from E17 ICR mouse embryos. Cerebral cortex tissues were rinsed with PBS, and incubated with 0.05% trypsin in PBS at 37 °C for 15 min. The cells were passed through a 70 μm cell strainer (22-363-548, Thermo Fisher Scientific, MA, USA), collected by centrifugation, and cultured in neurobasal medium (21103049, Thermo Fisher Scientific, Waltham, MA, USA) containing 2% B27 (17504044, Thermo Fisher Scientific, Waltham, MA, USA), 0.5 mM L-glutamine, and 1% Penicillin/Streptomycin (15140-122, Thermo Fisher Scientific, Waltham, MA, USA). Forty-eight hours later, the medium was changed to that containing 0.5 μM AraC (C3631, Sigma-Aldrich, St. Louis, MO, USA). On Day 4 of primary culture, we added 50 mM glutamate to culture medium for 3 h to induce apoptosis, or 25 μg/ml α-amanitin (A2263, Sigma-Aldrich, St. Louis, MO, USA) for 48 h to induce TRIAD necrosis. Neurons were then fixed with 4% FA and were subjected to immunohistochemistry. For western blot, we removed culture medium, added 100 μL sample buffer (62.5 mM Tris-HCl, pH 6.8, 2% SDS, 10% glycerol, 0.0025% BPB, 2.5% 2-mercaptoethanol) to each well of 12well plate, and recovered the samples. The samples were boiled for 10 min and subjected to SDS-PAGE.

**Ischemia induction of cerebral cortex tissues**. C57BL/6 J mice were anaesthetized with 1.0% Isoflurane® (099-06571, FUJIFILM, Osaka, Japan). Body temperature of the mice was maintained at 36.5 °C ± 0.5 °C during surgery with a heating plate. Skin and hair were disinfected with 70% ethyl alcohol, and a midline neck incision was made. The common carotid arteries were carefully dissected from fat tissues and the surrounding nerves not to injure vagal nerves, and pulled out with a surgical thread. After obtaining good view of the surgical field, bilateral common carotid arteries were clipped, using a microvascular clip (Dieffenbach Vessel Clip, straight 35 mm, Harvard Apparatus, Holliston, MA, USA) for 10 min. After the surgery, the mice were gently brought back to the cage, and watched carefully until recovered.

**Statistics**. Box plot is used to depict distribution of observed data, and the data are also plotted as dots. A box plot shows the median, quartiles and whiskers that represent 1.5× the interquartile range. In the other types of plots, values in each group are summarized by mean ± S.E.M.

Statistical differences between disease and control groups were evaluated by the Wilcoxon rank-sum test with post-hoc Bonferroni correction. Correlations between HMGB1 concentration and other markers in each individual subject were calculated using Pearson's correlation coefficient.

**Ethics**. This study was performed in strict accordance with the Guidelines for Proper Conduct of Animal Experiments by the Science Council of Japan. This study was approved by the Committees on Gene Recombination Experiments, Human Ethics, and Animal Experiments of the Tokyo Medical and Dental University (2016-007C6, O2014-005-09, and A2018-153A, respectively).

**Reporting summary**. Further information on research design is available in the Nature Research Reporting Summary linked to this article.

## Data availability

The authors declare that the data supporting the findings of this study are available within the article and supplementary information. Full anonymized data will be shared by request from any qualified investigator. The Source Data underlying Figs. 1a, b, 2b, c, e, g, h, 3b, c, 4b, 5b, c, d, e, 6b, c, e, g, i, j, 7b, d, e, f, 8c, 9b, c, d, e, f, g, h, 10c, d, e, h, i, j, k, l, n and o, and Supplementary Figs. 1, 2, 3, 4a, d, 6a, b, 9b, d, 10b, 11b, d, 12b, 14, 15a, b, c, d, e and f are provided as a Source Data file.

## Code availability

An original program code used to simulate a number of active cell death is available from our Website (http://suppl.atgc.info/041/).

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

## Acknowledgements

This work was supported by Brain Mapping by Integrated Neurotechnologies for Disease Studies (Brain/MINDS) from the Japan Agency for Medical Research and Development (AMED) (JP18dm0207013h0005); the Strategic Research Program for Brain Sciences (SRPBS) (JP18dm0107057h0002); and a Grant-in-Aid for Scientific Research on Innovative Areas "Foundation of Synapse and Neurocircuit Pathology" (22110001, 22110002) from the Ministry of Education, Culture, Sports, Science and Technology (MEXT) to H.O. This work is also partially supported by Strategic Research Program for Brain Sciences (SRPBS) (JP17dm0107057h0002) to K.T., brain bank supported by AMED (JP18dm0107103) to Y.S., and NIH grant (PO1AG14449) to E.M. We thank Marie Tanaka, Tayoko Tajima, and Emiko Yamanishi (Neuropathology, TMDU), and Naomi Takino and Mika Ito (Jichi Medical University), for technical support. We also thank Prof. Yutaka Hata, (Medical Biochemistry, TMDU) for pLL3.7-ires-GFP-TEAD-responsive-H2B-mCherry and RIKEN BioResource Center for providing APP-KI mice.

## Author contributions

H.T., K.K., K.F.: acquisition of data, drafting of manuscript, H.H.: analysis of data, drafting of manuscript. S.Y., X.J.: acquisition of data. M.W.: collection of samples, acquisition and analysis of data, drafting of manuscript. K.T.: acquisition of data. G.O.: collection of samples. A.I.: acquisition and analysis of data. N.A.: collection of samples. M.K., N.T., K.F., E.M.: collection of samples. H.A.: collection of samples, acquisition and analysis of data. T.S. and T.S.: collaboration on the mouse model. M.S.: discussion and suggestion for manuscript. Si.M.: collaboration on AAV. S.S., H.Okano., G.S., Y.S., S.M.: collection of samples, acquisition and analysis of data. H.Okazawa.: conception and design of the study, arrangement of collaboration, obtaining funding, drafting of manuscript.

## Competing interests

Shingo Yamada is an employee of Shino-Test Corporation. Hitoshi Okazawa and all other authors have nothing to report.
