## [Peer Review File · Nature Communications]

Reviewers' comments:

Reviewer #1 (Remarks to the Author):

This is a potentially interested manuscript that tackles a fundamental question in the AD field. In general, the data are strong and support the conclusions drawn by the authors. However, several issues decrease my overall enthusiasm for this manuscript.

1. The abstract should be edited as it is hard to follow due to a large number of undefined abbreviations.
2. Figure 1a-b does not add anything to the manuscript and should be removed. A better representation of the same data is shown in Fig. 1c-d.
3. The authors state "Presumably, the rate of decline in MMSE score, rather than the MMSE score itself, is correlated with the levels of CSF-HMGB1". However, they do not provide any scientific evidence to support this statement.
4. The authors need to validate their pSer46-MARCKS antibody with proper positive and negative controls.
5. The number of cases in Fig. 2d is low and as such, it is hard to make strong conclusions from what is presented. The same issue applies to Fig. 4b.
6. The rescue experiments of the Hippo pathway are very impressive. However, the authors should provide direct evidence of how rescuing this pathway decreases extracellular A β plaques.
7. Fig. 5: The changes in A β levels should be confirmed by sandwich ELISA, as quantification of IHC is often unreliable.

Reviewer #2 (Remarks to the Author):

The current study by Tanaka and coworkers was stemmed from their previous observation. They found that the HMGB1 level was significantly increased in the cerebrospinal fluid (CSF) of patients with mild cognitive impairment (MCI) but not Alzheimer's disease (AD). This abnormal change was associated with necrosis at the MCI stage. Using murine models and AD-iPSC derived neurons, they found that the increased necrosis was Hippo-pathway dependent and was mediated by YAP. The authors further discussed the potential usage of the Hippo pathway as a therapeutic target for AD. Overall, this is an interesting study, with ample data to support their conclusion. Although YAP had been proposed to be an important hub molecule in the regulatory network of AD (c.f. Ref. 6 in the text), the current study provided convincing data to further uncover the underlying mechanism of this molecule.

The authors might provide more data to confirm the direct link between the YAP deprivation and increased level of HMGB1 level at the MCI stage. Their current data are suggestive, and they may design a specific assay to show that YAP directly regulates the HMGB1 level via modulating necrosis. They should discuss this issue in the Discussion section.

It was shown that administration of sphingosine-1-phosphate (S1P) and YAPdeltaC61 (a neuron-specific isoform of YAP) rescued ER instability, necrosis, and A β pathology in AD mouse model. The authors might add some assays to further clarify the potential relationship between necrosis and A β pathology. If they could not supply new data this time, they should discuss this as a potential limitation of this study.

Minor issues:

1. The authors should cite related references in the first paragraph. This paragraph did not cite any references.
2. Lines 135-138 and extended data Figure 1, the authors may consider to include a test for amyloid- β precursor protein (APP)₆₆₉₋₇₁₁/amyloid- β (A β)₁₋₄₂ and A β ₁₋₄₀/A β ₁₋₄₂ ratios (Nakamura et al. 2018. Nature 554: 249-254).
3. Line 262, the author may present "data not shown" for the LATS1 action in neurons as an extended data file.

Reviewer #3 (Remarks to the Author):

extends previous studies by the Okazawa group, further developing the idea that deficiency of YAP1-mediated neuronal necrosis is increased at the pre-symptomatic stage of AD. Moreover, it represents a target for AD therapeutics since restoration of YAP1 levels via modulation of the hippo pathway or AAV-mediated YAP1 replacement rescued cognition and reduced abeta aggregation in AD models. These are important and novel aspects of the study. Additional findings related to early ER disruption and HMGB1 release overlap somewhat with the authors' previous reports but represent a worthwhile attempt to integrate all of the group's findings into a comprehensive hypothesis of AD pathological evolution, originating from intracellular abeta accumulation and early necrotic cell death leading to amyloidosis and plaque development. The general formulation proposed is supported by growing evidence that intracellular APP metabolite accumulations are the catalyst for AD development rather than the hallmark extracellular AD pathology. However, the details of the molecular sequence of events proposed require better documentation in some cases and various apparent inconsistencies need to be explained, as discussed below.

1. A major concern is the weakness of the evidence that links the various changes in putative necrosis-related indices within the same neuron or depicted population of neurons. A given change, such as YAP1 loss, ER swelling, LATs increase, etc are demonstrated, each by itself, and it is unclear whether the complete constellation of abnormalities that define this cell death process invariably co-exist in the same cell and whether the TRIAD pattern is the only degenerative pattern occurring in the brain or if other patterns (eg. necroptosis previously reported by others) are seen in some neurons. The notion of "active necrosis" is rather vaguely defined in the report and its features in some respects are not typical of previous descriptions of cell death in AD. For example, the enormous swelling implied by the ER tracker (and depicted by the authors in their previous reports of other disease) needs to be independently confirmed by EM and ER markers at a population level to clarify what this pattern looks like by conventional methods, how representative it is in AD brain, and why this pattern has not been commonly described in earlier AD studies.

2. The dying cell counting methodology is not well described and the identification criteria (or criterion) appears to be limited to DAPI labeling/nuclear morphology, which may be insufficient to distinguish the TRIAD form of necrosis from other closely related patterns. For example, the authors state that "Active necrosis is characterized by specific features of the nucleus" revealed by DAPI (see Fig. 2 legend), and the criteria used to describe the changes at 3 months are "single condensed DAPI region exists in the center of the nucleus, and is not accompanied by surrounding DAPI staining in nucleoplasm or the nuclear periphery (arrow)." By this way, should the small condensed DAPI signal appearing in the "1 month" 5XFAD be also considered as "active necrosis"? Another criterion, e.g. for 6 months, is "multiple nuclei aggregate to form a larger complex (arrows)". However, it is not clear if they counted the large DAPI signal in 6 month 5XFAD as 1 and the DAPI signal in 6 month APP-KI as 1 or 3. It is also not clear whether the authors use DAPI features or pSer46MARCKS signal for the counting of "active necrosis" in the MCI samples (Fig. 3c).

3. Another consideration in questioning the criteria using DAPI signal is that neurons showing depleted YAP signal (therefore implying that they are undergoing Hippo-pathway dependent necrosis) do not necessarily show the aforementioned DAPI features (Fig. 4a for MCI and Extended Data Fig 8a, c for 5XFAD and APP-KI). It would be expected that such cells would show both the DAPI changes and YAP changes. This means that even though the authors have tried to imply a link between changes in HMGB1/pSer46MARCKS and changes in the Hippo-pathway to make a coherent story for (non-necroptotic) necrosis, such a link is not fully convincing.

4. The authors also use pSer46-MARCKS as a necrosis marker and report an age-dependent decrease in 5xFAD mice; however, it looks like staining is age dependently increased in Fig2a, which needs to be explained. In terms of APP-KI mice, pSer46-MARCKS was ubiquitously upregulated at 1-month old: however, with maturation (i.e. 3-month old) only one neuron stained by pSer46-MARCKS, whereas adjacent neurons are negative for this phospho antibody. What is the explanation? Again, the AD occipital lobe stained by pSer46-MARCKS looks similar or more intense; however, the number of cells undergoing active necrosis is reported as decreased.

5. A number of considerations make the claim of selectivity for the YAP1-dependent pattern in MCI but not symptomatic AD puzzling and not easily explained. (a) First, the death of cell populations in a given region of AD is considered to be progressive and therefore at every stage of disease, some neurons are in the early stages of degeneration and others more advanced. The authors' formulation of AD events needs to explain why at more advanced stages of disease, there is not continued YAP1 dependent cell death or necrosis of a type that would release HMGB1 into the CSF at elevated levels. Is there a specific population dying early but not later? (b) it is not clear how it can be concluded that necrosis is declining based on the evidence presented in Fig2b. How does one distinguish the absence of a DAPI label due to advanced nuclear degradation in an "old" plaque from absence of a necrotic event (and therefore some other uncharacterized process) as the cause of the plaque? (c) The correlation between Braak staging and symptom emergence is not very reliable – MCI brains can have quite advanced pathology (eg. amyloid plaque depicted in MCI brain in Fig 2c), making the distinction between the pathology of MCI and AD blurred. Compounding the problem in this report is that the average MMSE for the symptomatic AD group overlaps with that of the MCI group and is not very different (22 vs 18). Moreover, there is no detailed description of the clinical staging of individuals in the symptomatic AD group to determine if there is any correlation between CSF-HMGB1 level and AD stage.

6. It needs to be explained why CSF-HMGB1 level decrease during disease progression. The authors state that CSF was collected from 48 control (34 normal. 14 disease control), 26 MCI and 73 AD patients. However only subset of sample was analysed (i.e. Fig 1: 30 control, 21 MCI, 56 AD patients ; Ext Fig1: 35 AD patients for Abeta and 18 AD patients for pTau study). Is there any reason for this selectivity for a subset or basis for eliminating samples, if this was done after analysis?

Minor points

1. need to provide a low-mag of Figure 3 or at least illustrate the boundary of neuron.
2. YAP co-staining with ER marker in Fig 4 MCI and AD sample.
3. Explain what is below the 75kDa band seen in Fig4 C, bottom WB.

Reviewers' comments:

Reviewer #1 (Remarks to the Author):

This is a potentially interested manuscript that tackles a fundamental question in the AD field. In general, the data are strong and support the conclusions drawn by the authors. However, several issues decrease my overall enthusiasm for this manuscript.

>> We appreciate very much kind and thoughtful evaluation of the reviewer.

1. The abstract should be edited as it is hard to follow due to a large number of undefined abbreviations.

>> Following the comment, we defined the abbreviations in the abstract.

2. Figure 1a-b does not add anything to the manuscript and should be removed. A better representation of the same data is shown in Fig. 1c-d.

>> Following the comment, we removed Figure 1a, b.

3. The authors state “Presumably, the rate of decline in MMSE score, rather than the MMSE score itself, is correlated with the levels of CSF-HMGB1”. However, they do not provide any scientific evidence to support this statement.

>> Following the comment, we removed the uncertain interpretation.

4. The authors need to validate their pSer46-MARCKS antibody with proper positive and negative controls.

>> We checked the property of anti-pSer46-MARCKS antibody in our paper published previously (Fujita et al, Sci Rep 2016, Sup Fig 2, as shown below).

As the reviewer pointed out, we had included the negative control in some lanes, but we had not performed experiments with positive control. We had just showed the different inhibitory effects of phospho-peptide (14 a.a. peptide including pSer46) and non-phospho-peptide on the interaction of the antibody with EGFP-MARCKS.

Supplementary Figure 2

In this revision, we performed ELISA assay showing the differential interaction of our anti-pSer46-MARCKS antibody with immobilized phospho- and non-phospho peptides matching to the 14 amino acids of MARCKS including Ser46 (new Extended Data Figure 4).

Moreover, to support the specificity of anti-pSer46-MARCKS antibody, we added immunohistochemistry of another antibody against non-phosphorylated MARCKS that was generated during the purification of anti-pSer46-MARCKS

antibody, and compared the results in immunohistochemistry of 5xFAD mice between non-phospho-Ser46 antibody and phospho-Ser46 antibody (new Extended Data Figure 4).

All these data support the predominant interaction of anti-pSer46-MARCKS antibody to pSer46-MARCKS.

5. The number of cases in Fig. 2d is low and as such, it is hard to make strong conclusions from what is presented. The same issue applies to Fig. 4b.

>> Following the comment, we increased the number of cases as possible as we can (Fig. 2d→new Fig. 2e; Fig. 4b→new Fig. 5b).

We also employed more strict definition of active necrosis in this revision than in previous version, and re-counted the number of active necrosis. Consequently, the number in AD was more decreased, and the difference between the two groups became more significant (Figure 2e).

We would accept the criticism that the data might still not be complete, but we think it can sufficiently suggest the tendency that active necrosis is highest in the early stage. We would like to point out that the number of cases is equivalent to a previous publication in Nature Neuroscience (Caccamo et al, Nat Neurosci 2017).

6. The rescue experiments of the Hippo pathway are very impressive. However, the authors should provide direct evidence of how rescuing this pathway decreases extracellular A β plaques.

>> We appreciate the critical comment from the reviewer. We firstly confirmed the decrease of extracellular A β by other additional experiments, i.e. WB and ELISA (new Figure 9e, f).

Regarding the mechanism, we added new results in new Figure 6 - 8 to reveal how intracellular A β triggers the ER ballooning cell death (TRIAD). Our new data in live imaging indicated that intracellular A β sequester YAP to its cytoplasmic aggregates, down-regulates YAP transcriptional activity, then YAP functional loss induces ER ballooning and cell death.

We had also showed in previous Extended Data Figure 10 (new Extended Data Figure 12) that intracellular A β remained in extracellular space as a deposit after the ballooning cell death, which might grew extracellular A β aggregates after the cell death.

For readers to understand our idea easily, we summarized our hypothesis in new Extended Data Figure 18.

7. Fig. 5: The changes in A β levels should be confirmed by sandwich ELISA, as quantification of IHC is often unreliable.

>> As aforementioned, we performed ELISA and western blot to confirm the decrease of A β levels by the treatments (new Figure 9e, f).

Reviewer #2 (Remarks to the Author):

The current study by Tanaka and coworkers was stemmed from their previous observation. They found that the HMGB1 level was significantly increased in the cerebrospinal fluid (CSF) of patients with mild cognitive impairment (MCI) but not Alzheimer's disease (AD). This abnormal change was associated with necrosis at the MCI stage. Using murine models and AD-iPSC derived neurons, they found that the increased necrosis was Hippo-pathway dependent and was medicated by YAP. The authors further discussed the potential usage of the Hippo pathway as a therapeutic target for AD. Overall, this is an interesting study, with ample data to support their conclusion. Although YAP had been proposed

to be an important hub molecule in the regulatory network of AD (c.f. Ref. 6 in the text), the current study provided convincing data to further uncover the underlying mechanism of this molecule.

>>> Thank you very much for the thoughtful and kind evaluation.

The authors might provide more data to confirm the direct link between the YAP deprivation and increased level of HMGB1 level at the MCI stage. Their current data are suggestive, and they may design a specific assay to show that YAP directly regulates the HMGB1 level via modulating necrosis. They should discuss this issue in the Discussion section.

>>> We appreciate the suggestion to show the direct link between deprivation of YAP and increased increase of HMGB1. First, we performed knockdown of YAP in normal iPSC-derived neurons by YAP-siRNA (new Figure 6f, g, h, i) and found it to increase specifically the ballooning necrosis, which was not induced by control siRNA. Necrosis should increase the extracellular HMGB1 according to a previous Nature paper by Dr. Bianchi's group (ref 23), and we directly quantified HMGB1 in culture medium after siRNA transfection by ELISA and confirmed the increased release of HMGB1 (new Figure 6j).

It was shown that administration of sphingosine-1-phosphate (S1P) and YAPdeltaC61 (a neuron-specific isoform of YAP) rescued ER instability, necrosis, and A β pathology in AD mouse model. The authors might add some assays to further clarify the potential relationship between necrosis and A β pathology. If they could not supply new data this time, they should discuss this as a potential limitation of this study.

>>> We appreciate the very critical comment from the reviewer. We added new data (new Figure 6 - 8) to show how intracellular A β triggers the ER ballooning cell death (TRIAD), i.e. intracellular A β simultaneously down-regulates YAP transcriptional activity, and then YAP functional loss induces ER ballooning cell death. The sequential processes were observed in a limited number of neurons

(new Figure 6) as well as at a single cell level (new Figure 8). We also performed YAP-knockdown in vivo, by using normal mice, and obtained data that support in vitro data (new Figure 7). Summarizing these results, we presented a hypothetical scheme deduced from this study (new Extended Data Figure 18).

Minor issues:

1. The authors should cite related references in the first paragraph. This paragraph did not cite any references.

>>> We added some representative references since there are numerous information about the failure of clinical trials of Abeta antibody and secretase inhibitors.

2. Lines 135-138 and extended data Figure 1, the authors may consider to include a test for amyloid- β precursor protein (APP)669-711/amyloid- β (A β)1-42 and A β 1-40/A β 1-42 ratios (Nakamura et al. 2018. Nature 554: 249-254).

>>> Following the advice, we added data regarding APP/ A β and A β 1-40/A β 1-42 ratios in new Extended Data Figure 1. Though the number of patients and controls were limited because data acquisition is limited for unnecessary examinations in clinical hospitals, the results were supportive for the previous publication in Nature (ref 25).

3. Line 262, the author may present “data not shown” for the LATS1 action in neurons as an extended data file.

>>> We added the data in new Extended Data Figure 8c and 8d.

Reviewer #3 (Remarks to the Author):

extends previous studies by the Okazawa group, further developing the idea that deficiency of YAP1- mediated neuronal necrosis is increased at the pre-symptomatic stage of AD. Moreover, it represents a target for AD therapeutics since restoration of YAP1 levels via modulation of the hippo pathway or AAV-mediated YAP1 replacement rescued cognition and reduced abeta aggregation in AD models. These are important and novel aspects of the study. Additional findings related to early ER disruption and HMGB1 release overlap somewhat with the authors' previous reports but represent a worthwhile attempt to integrate all of the group's findings into a comprehensive hypothesis of AD pathological evolution, originating from intracellular abeta accumulation and early necrotic cell death leading to amyloidosis and plaque development. The general formulation proposed is supported by growing evidence that intracellular APP metabolite accumulations are the catalyst for AD development rather than the hallmark extracellular AD pathology. However, the details of the molecular sequence of events proposed require better documentation in some cases and various apparent inconsistencies need to be explained, as discussed below.

1. A major concern is the weakness of the evidence that links the various changes in putative necrosis-related indices within the same neuron or depicted population of neurons. A given change, such as YAP1 loss, ER swelling, LATs increase, etc are demonstrated, each by itself, and it is unclear whether the complete constellation of abnormalities that define this cell death process invariably co-exist in the same cell

>>> We appreciate very much the critical comment from the reviewer. To respond to the concern, we performed new experiments with a single iPSC-derived neuron (new Figure 6, 8), in which we confirmed that intracellular accumulation of Amyloid beta sequesters YAP to cytoplasmic aggregates, which is followed by ER ballooning. Live imaging of a limited number as well as a

single neuron in these experiments elucidates the sequential events of Abeta accumulation → decrease of YAP activity → ER ballooning.

>>> LATS activation, as shown in the scheme of Hippo pathway below, is not essential, but a kind of permissive condition, for YAP-deprivation-induced cell death. YAP is the most downstream molecule in the Hippo pathway and it was shown sequestered by Amyloid beta from nucleus to cytoplasm within cells.

(Zhao et al, *Genes & Development* 2010. 24:862-874)

and whether the TRIAD pattern is the only degenerative pattern occurring in the brain or if other patterns (eg. necroptosis previously reported by others) are seen in some neurons.

>>> As the reviewer pointed out, a previous paper (Caccamo et al, Nat Neurosci 2017, ref 33) claimed that necroptosis occurs in the late phase of AD. Unfortunately, most of their data depend on antibodies against non-phosphorylated RIP1/3. Important issue is that they did not confirm co-localization of phospho-RIP1/3 and phospho MLKL in a single cell that is essential to transduce necroptosis signaling.

Therefore, we performed IHC of pRIP1/3 and pMLKL with AD model mouse and human AD samples. In this experiment, we did not detect any co-localized positive signals (pRIP1+pMLKL or pRIP3+pMLKL) in a single neuron in AD model mouse samples nor in human AD brains (new Extended Data Figure 16).

The notion of “active necrosis” is rather vaguely defined in the report

>> Active necrosis was strictly defined in new Figure 2a and described in the corresponding part of text.

and its features in some respects are not typical of previous descriptions of cell death in AD. For example, the enormous swelling implied by the ER tracker (and depicted by the authors in their previous reports of other disease) needs to be independently confirmed by EM and ER markers at a population level to clarify what this pattern looks like by conventional methods, how representative it is in AD brain,

>>> Following the comment, we performed EM (new Figure 4) and ER marker staining (Figure 4c, 8d) of MCI/AD patients. The images clearly demonstrated the ballooning of ER.

In addition, though the molecular analysis was not performed in this type of necrosis, multiple papers described similar morphological changes in human brains of AD and other neurodegenerative diseases, as listed below.

and why this pattern has not been commonly described in earlier AD studies.

>> If we carefully review previous works, similar or homologous cell death with cytoplasmic changes in human AD brains has been repetitively described in old historical papers of neuropathology. For instance, Hirano et al (J Neuropathol Exp Neurol 1968) reported granulovacuolar body, which is a homologous large vacuole found in pyramidal neurons in Sommer's sector of senile dementia, AD and Pick's disease (now a form of FTLD). Another examples is the paper by DW Dickson and colleagues (Act Neuropathol 1986), which described ballooned neurons in AD, Pick's disease, cortico-nigral degeneration, and pigment-spheroid degeneration. In our previous paper (Yamanishi et al, Act Neuropathol Commun 2017), we referred other papers describing homologous changes (Rebeiz et al, Arch Neurol 1968; Gibb et al, Brain 1989; Lippa et al, Hum Pthol 1990; Lowe et al, Neuropathol Appl Neurobiol 1992; Mori & Oda, Neuropathol 1997; Sakurai et al, Acta Neuropathol 2000). All these papers were now referred in this new revised version (ref 45 – 53).

The major advance in this paper is that we used pSer46-MARCKS as a marker of necrosis, which enabled us to detect such ballooning type of necrosis very easily. This is the reason why we could detect this kind of necrosis even though we are not so experienced as such old big neuropathologists.

2. The dying cell counting methodology is not well described and the identification criteria (or criterion) appears to be limited to DAPI labeling/nuclear morphology, which may be insufficient to distinguish the TRIAD form of necrosis from other closely related patterns. For example, the authors state that "Active necrosis is characterized by specific features of the nucleus" revealed by DAPI (see Fig. 2 legend), and the criteria used to describe the changes at 3 months are "single condensed DAPI region exists in the center of the nucleus, and is not accompanied by surrounding DAPI staining in nucleoplasm or the nuclear periphery (arrow)." By this way, should the small condensed DAPI signal appearing in the "1 month" 5XFAD be also considered as "active necrosis"? Another criterion, e.g. for 6 months, is "multiple nuclei aggregate to form a larger

complex (arrows)". However, it is not clear if they counted the large DAPI signal in 6 month 5XFAD as 1 and the DAPI signal in 6 month APP-KI as 1 or 3. It is also not clear whether the authors use DAPI features or pSer46MARCKS signal for the counting of "active necrosis" in the MCI samples (Fig. 3c)).

>> We really appreciate the critical comment. Although we had set the definition in our group, as the reviewer pointed out, we could not completely exclude the possibility that the examiner (KF) was a little bit confused and selected such non-representative images.

Therefore, we again shared the definition of "active necrosis" among lab members as described in the text, and also excluded decisively such confusion about the cluster of necrotic cells, by giving them another criteria "secondary necrosis". Furthermore, we added the third category "ghost of cell death", which seems to be the final state of sequential changes in mouse brains. All these categories are shown in Figure 2a in addition to the definition in the text.

KF, HT, KT and HO re-counted the cell death in images, made a consensus of counted number of active necrosis and others, and made new graphs (Figure 2b, c).

3. Another consideration in questioning the criteria using DAPI signal is that neurons showing depleted YAP signal (therefore implying that they are undergoing Hippo-pathway dependent necrosis) do not necessarily show the aforementioned DAPI features (Fig. 4a for MCI and Extended Data Fig 8a, c for 5XFAD and APP-KI). It would be expected that such cells would show both the DAPI changes and YAP changes.

>> As the reviewer commented, the previous images in Fig. 4a and Extended Data Fig 8a are inappropriate. The reason of these errors was described above, and we have replaced them with appropriate images in this revision.

As we revealed in new Figure 6 - 8, there is a series of molecular events in TRIAD under AD pathology. For instance, as shown in new Figure 8, YAP translocation to cytoplasm and ballooned ER is accompanied with nuclear shrinkage. DAPI or NucRed stains become weaker during and after shrinkage. In addition, ER rupture and release of HMGB1 triggering pSer46-MARCKS precede the nuclear shrinkage. Therefore, the definition using pSer46-MARCKS and DAPI has a sufficient rational. Simultaneously, the definition using pSer46-MARCKS and DAPI stains is the simplest way for in vivo analysis because YAP is released from cells as shown in Figure 8b and 8d, and because detection of active necrosis is easier with positive-pSer46-MARCKS marker than negative-YAP-marker.

This means that even though the authors have tried to imply a link between changes in HMGB1/pSer46MARCKS and changes in the Hippo-pathway to make a coherent story for (non-necroptotic) necrosis, such a link is not fully convincing.

>> As described above, our new data indicated that both YAP and HMGB1 are released from neurons during TRIAD under AD (new Figure 8b, 8d and Figure 6j). We previously reported that HMGB1 triggers MARCKS phosphorylation at Ser46 (Fujita et al, Sci Rep 2016, ref 22). All our data summarized in Extended Data Figure 18 suggested the link between YAP and HMGB1/pSer46-MARCKS (each reason is indicated in Extended Data Figure 18).

4. The authors also use pSer46-MARCKS as a necrosis marker and report an age-dependent decrease in 5xFAD mice; however, it looks like staining is age dependently increased in Fig2a, which needs to be explained.

>> This is again misunderstanding. In mass analysis we used the ratio between the value in 5xFAD and the value in B6/SJL mice (Fujita et al, Sci Rep 2016, ref 22).

(Figure 1c of Fujita et al, Sci Rep 2016, ref 22)

pSer46-MARCKS is increased not only during AD progression but also during normal aging. Therefore the ratio was the maximum at 3 months of age, while the exact amount of pSer46-MARCKS in 5xFAD mice is increased during aging, consistently with the immunohistochemistry data.

To make the point clear, we added WB in this revision and showed chronological change of the pSer46-MARCKS ratio between 5xFAD and non-Tg sibling mice (Extended Data Figure 4d).

Consistently with our mass analysis result reported previously (Fujita et al, Sci Rep 2016, ref 22), the ratio in WB declined from 3 months of age though the value of pSer46-MARCKS in 5xFAD mice by itself was increased.

We would like to add that total pSer46-MARCKS signal in IHC could increase even though active necrosis (a single neuron necrosis accompanied with a small pSer46-MARCKS reaction) peaks at 2 or 3 months because the secondary necrosis (multiple neurons' necrosis accompanied with a large pSer46-MARCKS reaction) increases from 6 to 12 months, which should increase the total amount of pSer46-MARCKS in tissue.

In terms of APP-KI mice, pSer46-MARCKS was ubiquitously upregulated at 1-month old: however, with maturation (i.e. 3-month old) only one neuron stained by pSer46-MARCKS, whereas adjacent neurons are negative for this phospho antibody. What is the explanation?

>>> First, we replaced the previous figure 2a and 2b, because there were not suitable to show the precise concept of “active necrosis”. The images later than 6 months in Fig 2b should be categorized as “secondary necrosis” now.

Second, I am sorry but we are afraid that this comment is a confusion of the reviewer. pSer46-MARCKS is a response to necrosis and NOT necrosis by itself. If one observes carefully, one can understand that most of surrounding “cell bodies” are out side of band of pSer46-MARCKS. pSer46-MARCKS are degenerative neurites as we have shown in the previous paper (Fujita et al, Sci Rep 2016, ref 22) and as we show in immuno-EM in this work (new Extended Data Figure 5, also has shown in the previous version of this manuscript). At the center of pSer46-MARCKS band, a necrosis exists and HMGB1 released from the necrosis will not affect such cell bodies that are more distant than the pSer46-MARCKS band. I would like to show you a figure for understanding.

Again, the AD occipital lobe stained by pSer46-MARCKS looks similar or more intense; however, the number of cells undergoing active necrosis is reported as decreased.

>> This is the comment on previous Figure 2c. This is due to the tendency of a member (KF) who sometimes automatically shows the most impressive image for him as a representative image. This attitude was totally wrong and the previous image is not definitely representative. Multiple members (HT, KT, HO) checked and confirmed that an alternative image is better (new Figure 2d). We could only detect “secondary necrosis” in these patients, which was confusingly shown in the previous version, and there was no “active necrosis”. Counting was also newly performed and the graph was revised (new Figure 2b, c, e).

5. A number of considerations make the claim of selectivity for the YAP1-dependent pattern in MCI but not symptomatic AD puzzling and not easily explained.

>> This is misunderstanding. We never say that YAP-necrosis is specific to MCI and not symptomatic AD. We did claim that YAP-necrosis-related events continue in clinical AD state. In other words, YAP-necrosis continuously occurs from MCI to full AD state. However, the frequency of YAP-necrosis is highest in the early stage (MCI stage) and declines in late stage (symptomatic AD stage), simply. The reason will be described in the next page of this rebuttal.

As shown in Figure 2, the “ghost of cell death” pattern is increasing during the time course. This pattern, which is quite similar to senile plaque that we pathologist knows, does not accompany strong reaction of pSer46-MARCKS in surrounding cells. It means this structure is NOT actively releasing HMGB1.

So the “YAP-dependent pattern”, which is not ours but original term of the reviewer, corresponds to “active necrosis” in our definition, and it is decreased in advanced state of human AD. Also “secondary necrosis” that should also release HMGB1 should be decreased in advanced state of human AD.

(a) First, the death of cell populations in a given region of AD is considered to be progressive and therefore at every stage of disease, some neurons are in the early stages of degeneration and others more advanced. The authors’ formulation of AD events needs to explain why at more advanced stages of disease, there is not continued YAP1 dependent cell death or necrosis of a type that would release HMGB1 into the CSF at elevated levels. Is there a specific population dying early but not later?

>> This is obviously a mathematics problem. If total number of neurons at the initial time point = N , the residual number at current time = N_k

$$N_k = N (1 - r)^{k-1}$$

Time, which is the number of cell death cycles, is k. The rate of cell death is r.
Active cell death is

$$\text{Active cell death} = N_k - N_{k-1} = N r (1 - r)^{k-1}$$

In this mathematics, the real initial time point for cell death is k=0.

In this formulation, as the graph shows, active cell death initiates around 2 months that matches intracellular Abeta accumulation. Although examiners of the morphological evaluation in Figure 2 did not know this mathematical expectation, the result matches well with the formula!!

We can also speculate HMGB1, which is released from active cell death at each time point, and showed it in a graph.

We do not need to have a specific population of neurons in all the calculations, to explain the active necrosis number is declining during the course of disease,

We showed these results in Figure 2f, g and h.

We believe that readers can understand the reason of increase in secondary necrosis and ghost of cell death during the time course because they are subsequent process and ghost of cell death accumulates.

(b) it is not clear how it can be concluded that necrosis is declining based on the evidence presented in Fig2b.

>> This has been explained as above.

How does one distinguish the absence of a DAPI label due to advanced nuclear degradation in an "old" plaque from absence of a necrotic event (and therefore some other uncharacterized process) as the cause of the plaque?

>> We might not have exactly understood English, but this is what we have been imagining from the first version of the paper. DAPI should disappear after nuclear degradation. This category we named as “ghost of cell death”, which is usually recognized as senile plaque by neuropathologists.

If the reviewer means that necroptosis or undefined (unknown) pathology leads to amyloid plaques, we would answer as follows. First our data deny co-activation of RIP1/3 and MLKL in a single neuron in MCI/AD brains that is indispensable for necroptosis execution (new Extended Data Figure 16), excluding necroptosis. A previous report in Nat Neurosci suggested single activation of RIP1/3 or MLKL mainly using non-phospho- antibodies. However, this seems not the case. The third unknown pathology might induce similarly amyloid plaque finally. We could not exclude the possibility. But this is not our task. We could not exclude unknown thing. This is a general rule of scientific research that all scientists should follow. The demand is surprising.

(c) The correlation between Braak staging and symptom emergence is not very reliable – MCI brains can have quite advanced pathology (eg. amyloid plaque depicted in MCI brain in Fig 2c), making the distinction between the pathology of MCI and AD blurred.

>> We never have the opposite view to the reviewer, and instead we similarly consider that pathological and clinical stages are sometimes very discrepant. It is well known that people who have high IQ, for instance university professors, sometimes have extremely advanced AD pathology although he was not so demented by his death. Such stories are well known among experienced neurologist including me. It is a common sense that AD and MCI are overlapped in the aspect of human pathology.

But we are now discussing about the tendencies between MCI and AD groups in various aspects, and never discussing about the specific cases of MCI or AD. It is very natural and well-known to find a lot of amyloid plaques in MCI patients,

and if the reviewer said that the patient is not clinically MCI from that reason, it is quite surprising for clinical neurologists in general. NO-body will claim against that the AD group is “generally” more advanced as a population than MCI group in the aspect of the morphological and biochemical changes of the AD pathology.

Meanwhile, we think that we should describe Braak stage, so we added the information (page 30, new Figure 4b legend).

Compounding the problem in this report is that the average MMSE for the symptomatic AD group overlaps with that of the MCI group and is not very different (22 vs 18). Moreover, there is no detailed description of the clinical staging of individuals in the symptomatic AD group to determine if there is any correlation between CSF-HMGB1 level and AD stage.

>> This comment again sounds a little bit strange for clinical neurologists. It is very well-known among neurology specialists that MMSE scores overlap between MCI and AD groups, because the diagnostic criteria of MCI is based on his/her subjective claim for intellectual abnormalities and rather than the score in a set of clinical examinations, and the diagnosis is NOT based on the MMSE score. In diagnosis of AD dementia, objective confirmation of sustaining and progressive decline of intellectual ability is needed and therefore MMSE and other scores should be low. But in the case of MCI diagnosis, it is demanded to exclude obvious dementia but still the cases with low scores of MMSE are NOT excluded from MCI diagnosis according to the criteria of ICD10, DSM5 or NIA-AA.

In addition, we would like to stress that the diagnosis of patients is made in top-level university hospitals in Japan that are independent of us (TMDU), and the corresponding author believes in their diagnosis. We all co-authors wonder if the reviewer is not a well-experienced clinical neurologist who has managed a number of dementia and MCI patients.

We showed MMSE and ADS in Extended Data Table 2. As the CSF samples are given from other university hospitals we cannot claim that two of MCI patients did not receive MMSE or ADS-cog evaluation, but all AD patients and most MCI patients have received MMSE and/or ADS/FAB evaluation (Extended Data Table 2). Other detailed history in each patient is almost impossible to show. Brain biopsy from living patients is prohibited (at least in Japan), and we never know that the exact pathological AD stage of living patients who provided CSF samples.

6. It needs to be explained why CSF-HMGB1 level decrease during disease progression.

>> This is because the sum of “active necrosis” and “secondary necrosis” releasing HMGB1 declines at the late stage of AD (Figure 2).

The authors state that CSF was collected from 48 control (34 normal. 14 disease control), 26 MCI and 73 AD patients. However only subset of sample was analysed (i.e. Fig 1: 30 control, 21 MCI, 56 AD patients ; Ext Fig1: 35 AD patients for Abeta and 18 AD patients for pTau study). Is there any reason for this selectivity for a subset or basis for eliminating samples, if this was done after analysis?

>>> We excluded samples that are contaminated with blood or unsuitable in technical reasons (for example, the sample is stuck in Mass Spec).

Minor points

1. need to provide a low-mag of Figure 3 or at least illustrate the boundary of neuron.

>> Following indication, we illustrate the boundary of a neuron.

2. YAP co-staining with ER marker in Fig 4 MCI and AD sample.

>> Following indication, we performed the requested experiment (new Figure 8d).

3. Explain what is below the 75kDa band seen in Fig4 C, bottom WB.

>> We here paste Figure 4c in the previous version.

The reviewer said it is the bottom WB and the band is below 75kD, so we speculate that the reviewer is asking the band (indicated with white arrow), which is generally mentioned as the high molecular oligomer. A small aliquot seems leaked from lane 2 to lane 1. We indicated the HMW oligomer in the new figure.

Reviewers' comments:

Reviewer #1 (Remarks to the Author):

The authors have addressed my comments, successfully.

Reviewer #2 (Remarks to the Author):

The authors had properly answered my queries. I am happy with this revised version.

Reviewer #3 (Remarks to the Author):

The authors have made an effort to address the issues raised in the initial review, but there are still existing or new major concerns.

1. The authors now establish 3 groups to separate an "active necrosis" group from the other two groups (Fig. 2a) which helps an understanding of how the quantification in the original Fig 2a was made. We now know the definition of "active necrosis" as "single dying cell with reactive pSer46-MARCKS". However, these two criteria alone are unconvincing to identify "active necrosis" and, if adopted by others in the field would add further confusion to an already murky field. For example, identification of "single dying cell" is mainly based on the observation of some (residual) DAPI signal (Fig. 2a, b) but it can be argued that such a DAPI signal could be a result of any type of cell death. The authors may then argue that they also include something more specific to necrosis (in particular TRIAD), i.e., the "reactive pSer46-MARCKS", which itself is not a validated specific marker of necrosis. Moreover, as the authors mentioned in this paper and in their previous publications, the signal of reactive pSer46-MARCKS may mainly represent degenerative neurites in AD models used. Therefore, how can the authors exclude the possibility that these pSer46-MARCKS-positive neurites is not coming from the "single dying cell" implied by the residual DAPI signal but rather coming from passing neurites originating from other neurons? Actually, as stated in the paragraph starting at L216, "pSer46-MARCKS reactivity increased in neurons surrounding dying cells...", suggesting that the authors may realize that the "dying cells" and the "reactive pSer46-MARCKS" neurites could be components derived from separate neurons. Therefore, the authors have applied two criteria that individually are not shown to be specific for necrosis and together are insufficient to distinguish a necrotic pattern or distinguish death of a single cell from a possible contribution of neuritic degeneration from neighboring cells. Although EM images have been demonstrated in Extended Data Figure 5a, such ultrastructural information, even with what is shown in the new speculative depiction of the process diagrammed on P15 of the "Rebut" file, cannot resolve these concerns.

2. The authors have now explained the sources/diagnoses/neuropathology of the human samples and provided the staging info for MCI (Braak stage III) and AD (Braak stage V). For some of the reasons the authors discuss in their response, the continued use of 'MCI' to stage the postmortem brains rather than using pathological criteria is problematic. Most importantly, one of the key themes from this paper is that the specific "active necrosis" occurs predominantly in MCI, which begs the key questions as to how does, or can, the mechanism of cell death shift so dramatically from "active necrosis" at Braak stage III to another (uncharacterized) form of cell death at Braak stage V at which stage there is still considerable ongoing cell death. This implied shift in death mechanism is not scientifically intuitive and needs to be explained or demonstrated to support the main conclusion that there is a special cell death mode in MCI. Although MCI samples exhibited a huge number of "ER blooming cells" compared to the control, the number for AD is not significantly different from that for the control (Fig. 4b, graph). How can the authors be sure that the determination/quantification of "ER blooming" is valid, given that the electron lucent, swollen structures quantified could represent any number of different swollen organelle subtypes? In fact,

the very unusual single ballooned profiles in the neurons of the AD model are even more puzzling as to identity since they appear singly in the cell in the absence of any other swollen vesicles as would be expected in a conventional pattern of necrosis and they even look different from the multiple profiles depicted in the author's earlier report on TRIAD in Huntington disease. In the ICC demonstration in Fig4c, it is difficult to appreciate what the authors consider to be supporting evidence: the distinction between the depicted cells are not apparent and are not pointed out by the authors. At minimum, identity of the structures should be established by performing IEM with an ER marker to be sure that the "blooming" structures in the human and the mouse samples are really ER-related.

3. The authors agreed with the reviewer that the previous Fig. 4a and Extended Data Fig. 8a were "inappropriate" and "we have replaced them with appropriate images in this revision". However, they did not. The same figures have been kept and are now Fig. 5a and Extended Data Fig. 10a so the original concern about these data remains.

4. The necrosis under this study regards a relatively specific type of necrosis, either "transcriptional repression-induced atypical cell death (TRIAD)", "non-necroptotic" or "YAP-dependent". However, in many cases, the general term "necrosis" has been used. The use of term(s) should be consistent throughout the text if possible and should be clarified at the beginning of the report. Even so, the defining characteristics of necrosis used to identify necrosis in the report, as discussed above, are unconventional, not well validated, and insufficient to establish the mode of cell death in these experimental conditions.

Reviewers' comments:

Reviewer #1 (Remarks to the Author):

The authors have addressed my comments, successfully.

>>> We appreciate very much kind evaluation by Reviewer #1.

Reviewer #2 (Remarks to the Author):

The authors had properly answered my queries. I am happy with this revised version.

>>> We appreciate very much kind evaluation by Reviewer #2.

Reviewer #3 (Remarks to the Author):

The authors have made an effort to address the issues raised in the initial review, but there are still existing or new major concerns.

>>> We appreciate great efforts of Reviewer #3 for evaluation of our manuscript.

1. The authors now establish 3 groups to separate an “active necrosis” group from the other two groups (Fig. 2a) which helps an understanding of how the quantification in the original Fig 2a was made. We now know the definition of “active necrosis” as “single dying cell with reactive pSer46-MARCKS”. However, these two criteria alone are unconvincing to identify “active necrosis” and, if adopted by others in the field would add further confusion to an already murky field. For example, identification of “single dying cell” is mainly based on the observation of some (residual) DAPI signal (Fig. 2a, b) but it can be argued that such a DAPI signal could be a result of any type of cell death.

>>> Basically, the term “necrosis” is used on the basis of the morphological classification. The frequently referred classification (by Schweichel and Merker, 1973; Clarke 1990) categorized cell death into Type 1, 2 and 3. But type 2 and 3 are characterized by cytoplasmic changes such as organelle vacuolation without remarkable changes of nucleus, and therefore both are included in the concept of necrosis. Type 2 is vacuolation of autophagosome, while vacuoles of Type 3 are derived from lysosome or other organelle. Type 1 is apoptosis, of course.

The first claim from the reviewer is that the change of DAPI signal might occur in apoptosis, but apoptosis is characterized by the nuclear changes, i.e. chromatin condensation, and the condensed chromatin remain through the processes, and therefore DAPI signal in apoptotic cells remains as relatively high signals. This is the reason why a huge number of papers have used DAPI stain as a proof of apoptosis in their researches in vitro and in vivo. The following image is one example that condensed chromatin remain as apoptotic body in cell at an advanced stage of apoptosis (Okazawa et al, JCB 1996). Of course apoptosis could include cytoplasmic changes, but the nuclear chromatin condensation and shrinkage of cytoplasm are essential for apoptosis. Phagocytes rapidly remove such cells showing “eat me signal” in vivo.

Even if we accept the reviewer’s claim “the disappearance of DAPI signal might occur in apoptosis”, which go through the net of phagocytosis, as a rare case in vivo, reactive pSer46MARCKS in neurites of surrounding cells is specific to necrosis given that following evidences are reported and newly shown in this paper.

First, HMGB1 is released from cells to extracellular space under necrosis but not from cells under apoptosis as reported by Prof. Bianchi’s group (Scaffidi et al, Nature 2002). Second, extracellular HMGB1 triggers cell signaling after binding to TLR4 at the cell surface and leads to MARCKS phosphorylation at pSer46 (Fujita et al, Sci Rep 2015).

Third, by in vitro experiments of primary cortical neurons, we actually showed that TRIAD necrosis but not apoptosis induces pSer46MARCKS (Extended Data Figure 6). The difference of the reactive increase of pSer46MARCKS in surviving cells in neighborhood is so drastic between TRIAD necrosis and apoptosis.

>>> Especially, we regret the reviewer's comment "if adopted by others in the field would add further confusion to an already murky field". This is clearly intentional overstatement that inhibits the progress of science. The "murky field", if it really is, could never be solved without a new hypothesis, and the histories of numerous scientific fields have shown this immutable truth.

The authors may then argue that they also include something more specific to necrosis (in particular TRIAD), i.e., the "reactive pSer46-MARCKS", which itself is not a validated specific marker of necrosis.

>>> We raised three reasons as abovementioned, and "reactive pSer46-MARCKS", which itself is validated as a specific marker of necrosis. We have shown data that apoptotic cortical neurons do not induce pSer46-MARCKS in surrounding neuron or neurite while TRIAD necrosis dramatically induced pSer46-MARCKS, in this paper as "new Extended Data Figure 6".

Moreover, as the authors mentioned in this paper and in their previous publications, the signal of reactive pSer46-MARCKS may mainly represent degenerative neurites in AD models used. Therefore, how can the authors exclude the possibility that these pSer46-MARCKS-positive neurites is not coming from the "single dying cell" implied by the residual DAPI signal but rather coming from passing neurites originating from other neurons? Actually, as stated in the paragraph starting at L216, "pSer46-MARCKS reactivity increased in neurons surrounding dying cells...", suggesting that the authors may realize that the "dying cells" and the "reactive pSer46-MARCKS" neurites could be components derived from separate neurons.

>>> This is exactly what we meant. We believe that a dying cell is circumscribed by degenerative neurites derived from separate neurons.

Therefore, the authors have applied two criteria that individually are not shown to be specific for necrosis

>>> We applied two elements (as a complex) for the diagnosis of active necrosis, one is the feature of the dying cell by itself and the other is the feature of surrounding cells. As explained above, both are specific (even though not 100%) for necrosis.

and together are insufficient to distinguish a necrotic pattern

>>> As mentioned above, necrosis but not apoptosis could induce pSer46-MARCKS in surrounding cells. Also, DAPI should remain in apoptotic cells

or cells at advanced stages should be removed rapidly in vivo by phagocytes. It is sufficient to exclude apoptosis and indicates necrosis is responsible for the cell death we observed.

or distinguish death of a single cell from a possible contribution of neuritic degeneration from neighboring cells.

>>> It is hard to consider that neurite degeneration induces the cell death of a separate cell. For instance as shown in Ext Data Fig 5a, the death process of a neuron at the center is more advanced than the surrounding neurites of other neurons whose morphology is relatively preserved. Chronological relationship from surrounding neurites to a cell at the center in the assumption is opposite to the actual observation.

Although EM images have been demonstrated in Extended Data Figure 5a, such ultrastructural information, even with what is shown in the new speculative depiction of the process diagrammed on P15 of the “Rebut” file, cannot resolve these concerns.

>>> As we have explained, the model that the reviewer has suggested (or concerned) in his comments is similar to the model we have meant (at 15 page in Rebuttal letter in previous submission). We have been preparing another paper including other IEM images (pasted below). Such images obviously indicate that the degenerative neurite (gold particles reflect pSer46MARCKS antibody stains) is not coming from the neuron with a deforming and shrinking “non-apoptotic” nucleus (n). Since such images further supports our concept that a necrotic neuron induces reactive degeneration of neurites surrounding it, we decided to include the images in Extended Data Figure 5 (Extended Data Figure 5c, d).

2. The authors have now explained the sources/diagnoses/neuropathology of the human samples and provided the staging info for MCI (Braak stage III) and AD (Braak stage V). For some of the reasons the authors discuss in their response, the continued use of ‘MCI’ to stage the postmortem brains rather than using pathological criteria is problematic. Most importantly, one of the key themes from this paper is that the specific “active necrosis” occurs predominantly in MCI, which begs the key questions as to how does, or can, the mechanism of cell death shift so dramatically from “active necrosis” at Braak stage III to another (uncharacterized) form of cell death at Braak stage V at which stage there is still considerable ongoing cell death. This implied shift in death mechanism is not scientifically intuitive and needs to be explained or demonstrated to support the main conclusion that there is a special cell death mode in MCI.

>>> The reviewer seems to suggest that “active necrosis” is fundamentally distinct from cell death in the late pathological stage. But we have not shown by our hands that different type of cell death exists in the late stage. Instead we have shown the similar “active necrosis” in the late stage.

The reviewer might suspect that “active necrosis” is fundamentally distinct from “secondary necrosis”. However, “secondary necrosis” looks similar to “active necrosis” from the aspects of reactive increase of pSer46MARCKS in surrounding cells and of accumulation of amyloid beta. Though further investigation will be necessary in the future, and though chronological positions of “active necrosis” and “secondary necrosis” in the same set of grouped cell death are different, there is no reason to distinguish their molecular mechanisms.

Alternatively, the reviewer might imagine that the other types of cell death (such as necroptosis) actually exist in a later stage of AD. But we could not obtain the evidence that such other cell death types occur even in the late stage of AD as we had described in rebuttal and has shown in the manuscript **in previous submission**. Instead, our data were not supportive for necroptosis.

==== the comment of the reviewer and our rebuttal **in the previous submission** ====
.....and whether the TRIAD pattern is the only degenerative pattern occurring in the brain or if other patterns (eg. necroptosis previously reported by others) are seen in some neurons.

>>> *As the reviewer pointed out, a previous paper (Caccamo et al, Nat Neurosci 2017, ref 33) claimed that necroptosis occurs in the late phase of AD. Unfortunately, most of their data depend on antibodies against non-phosphorylated RIP1/3. Important issue is that they did not confirm co-localization of phospho-RIP1/3 and phospho MLKL in a single cell that is essential to transduce necroptosis signaling.*

Therefore, we performed IHC of pRIP1/3 and pMLKL with AD model mouse and human AD samples. In this experiment, we did not detect any co-localized positive signals (pRIP1+pMLKL or pRIP3+pMLKL) in a single neuron in AD model mouse samples nor in human AD brains (new Extended Data Figure 16).

===== end =====

>>>Therefore, our data did not indicate (or we did not claim) the shift of cell death characteristics during the course of AD progression.

However, our goal is, of course, not to deny the work published by others but to reveal that TRIAD cell death occurs in the early stage of AD.

Although MCI samples exhibited a huge number of “ER blooming cells” compared to the control, the number for AD is not significantly different from that for the control (Fig. 4b, graph). How can the authors be sure that the determination/quantification of “ER blooming” is valid, given that the electron lucent, swollen structures quantified could represent any number of different swollen organelle subtypes? In fact, the very unusual single ballooned profiles in the neurons of the AD model are even more puzzling as to identity since they appear singly in the cell in the absence of any other swollen vesicles as would be expected in a conventional pattern of necrosis and they even look different from the multiple profiles depicted in the author’s earlier report on TRIAD in Huntington disease. In the ICC demonstration in Fig4c, it is difficult to appreciate what the authors consider to be supporting evidence: the distinction between the depicted cells are not apparent and are not pointed out by the authors. At minimum, identity of the structures should be established by performing IEM with an ER marker to be sure that the “blooming” structures in the human and the mouse samples are really ER-related.

>>> The reviewer asked whether the ballooning (we do not say blooming) structure is derived from ER and whether the ballooning structure is single or multiple. For the first point, as most human pathologists know, postmortem human brain samples do not show ideal EM images. We agree with the opinion. But this is the reason why we need to analyze mouse models in parallel.

Regarding the number of vacuoles in cell (single or multiple), as the reviewer knows, ER is a connected lumen structure that looks separated to multiple structures in a sliced section image. In addition, during the process of ER ballooning, multiple parts of an ER lumen expand (we showed such chronological changes of ER in Supplementary Videos of this paper and our previous paper). Combining these two notions, it is natural to consider that a single large vacuole (when it looks like a

single) is the terminal stage of ballooning of ER and that multiple large vacuoles are under the process reaching to the terminal stage. Therefore, these states, although that look different at a glance, indicate the identical pathological process.

Regarding the origin of vacuole, we added in Figure 4 a new image of intermediate size of ERs (new Figure 4c) showing that ballooned ER accompanies ribosome (the definite proof of rough ER) at an intermediate size but loses ribosome when it expands to larger sizes. Such larger ER still kept on the surface a small number of ribosomes even in human MCI samples (new Figure 4c), which aided our identification of the origin. A very few residual ribosomes are detected in huge vacuoles (Figure 4b), supporting that they were originated from ER.

Immuno-staining of the ER content with anti-KDEL antibody also confirmed the ballooning of ER in the cerebral cortex neurons of 5xFAD mice before onset (3 months of age) and of human MCI patients (Figure 4d, e). Responding to the comment of the reviewer that the calnexin immunostains were difficult to show such ballooning ER, we reduced the number of panels showing calnexin immunostains and mainly used KEDL stain images to support the conclusion. We also added arrows in the panels to point out the ER ballooning.

In our previous paper (Hoshino et al, JCB 2006), we had also proved that ER is ballooned in TRIAD by using the ER-specific marker, and showed by EM that extremely expanded ERs lose ribosome. Therefore, extreme large vacuoles without ribosomes observed in this study do not contradict our conclusion, but instead supporting our conclusion that vacuoles are derived from ER.

3. The authors agreed with the reviewer that the previous Fig. 4a and Extended Data Fig. 8a were “inappropriate” and “we have replaced them with appropriate images in this revision”. However, they did not. The same figures have been kept and are now Fig. 5a and Extended Data Fig. 10a so the original concern about these data remains.

>>> Regarding previous Fig. 4a (Fig 5a in R2 version) and Extended Data Fig. 8a (Extended Data Fig. 11a in R2 version), they were appropriate for showing colocalization of YAP and Abeta, but we needed to pay more attention to DAPI signal intensities in the images. In this sense, we appreciate very much about the criticism of the reviewer. In this revision, the DAPI signals were normalized and adjusted among multiple panels by using DAPI signals of glia cells, and the old panels were replaced with newly corrected ones at the same visual fields.

Together with the new corrected images, when we get back to the reviewer's original question, we should better answer as follows.

===== original comment from the reviewer =====

3. *Another consideration in questioning the criteria using DAPI signal is that neurons showing depleted YAP signal (therefore implying that they are undergoing Hippo-pathway dependent necrosis) do not necessarily show the aforementioned DAPI features (Fig. 4a for MCI and Extended Data Fig 8a, c for 5XFAD and APP-KI). It would be expected that such cells would show both the DAPI changes and YAP changes.*

===== end =====

As described above, we normalized signal intensities of DAPI stains among image panels using the DAPI stains of glia, and showed DAPI-only images in Fig 5a in R2 version (previous Fig. 4a) and Extended Data Fig. 11a in R2 version (previous Extended Data Fig. 8a). The new DAPI-only images revealed the decrease of DAPI in neurons possessing cytoplasmic YAP and Abeta, even though the decrease was not so drastic as in the final stage of cell death (“active cell death”) shown in Figure 2.

In addition, we quantified the DAPI signals, and compared them in the graphs between amyloid+ / cytoplasmic YAP+ neurons and amyloid- / cytoplasmic YAP- neurons in MCI patients (Fig 5b) and in mouse AD models (Extended Data Fig. 11b, d). Again, it was shown that DAPI was decreased in neurons possessing cytoplasmic YAP and Abeta.

However, the reviewer might feel that DAPI stains in Figure 2 of “active necrosis” were further weak. For this issue, we think as follows. From the observation in Figure 8, nuclear stains (in this case we used NucRed instead of DAPI) remained at least for 28 hours after the shift of YAP from nucleus to cytoplasm. This indicates that there is a long time lag even between YAP change and DAPI/NucRed change in the case of *in vitro* cortical neurons. In the case of *in vivo* cortical neurons, there would be many environmental and/or niche mechanisms to support and prevent neurons from death such as glia, adhesion molecules and trophic factors. They would further delay the cell death process and elongate the time lag.

In Fig 5a (previous Fig. 4a) and Extended Data Fig. 11a (previous Extended Data Fig. 8a) in R2 version, we used cytoplasmic Abeta and were not using pSer46MARCKS. During the course of TRIAD in AD, the phase detected by intracellular Abeta was at the early stage while the phase detected by pSer46MARCKS in surrounding cells was at the terminal. Such a weak DAPI at the terminal stage, when YAP signal has already disappeared in the dying cells (please

have a look at the +14hr image of Figure 8b), could not be detected without using pSer46MARCKS in surrounding cells.

4. The necrosis under this study regards a relatively specific type of necrosis, either “transcriptional repression-induced atypical cell death (TRIAD)”, “non-necroptotic” or “YAP-dependent”. However, in many cases, the general term “necrosis” has been used. The use of term(s) should be consistent throughout the text if possible and should be clarified at the beginning of the report. Even so, the defining characteristics of necrosis used to identify necrosis in the report, as discussed above, are unconventional, not well validated, and insufficient to establish the mode of cell death in these experimental conditions.

>>> We used different terms from a wide definition to a narrow (more strict) definition according to the progression of each experiments. In most papers handling the previously defined concept, the authors use the same term from the beginning of the paper. It is possible to use TRIAD if it is definite in Figure 1, but as the reviewer commented it is not the case. We changed the term according to the deepening of our comprehension of the characteristics of our cell death, and we think this is more honest way.

We reported “transcriptional repression-induced atypical cell death (TRIAD)” that is classified to Type 3B cell death and caused by alpha-amanitin (the RNA polymerase II specific inhibitor proven by structural analysis of the Nobel prize winner, Prof. Roger Kornberg) in neuron, and showed that TRIAD is related to YAP (Hoshino et al, JCB 2006). Later, we also confirmed that YAP regulates TRIAD and that YAP-dependency is the hallmark of TRIAD (Mao et al, Hum Mol Genet 2016; Mao et al, CDDis 2016). In this study, cell death of AD had the defined hallmark of TRIAD, and it is also “non-necroptotic” judging from the marker of necroptosis as described above.

Reviewers' comments:

Reviewer #3 (Remarks to the Author):

The authors have made significant efforts to address the remaining concerns, including a new Extended Data Figure 6 of neuronal cultures showing that "necrosis but not apoptosis could induce pSer46-MARCKS in surrounding cells" which supports their view that pSer46-MARCKS is a specific marker for necrosis when combined with the DAPI signal for identifying the "active necrosis". They made their classification of cell death simpler by citing Schweichel and Merker, 1973; Clarke 1990 to just mention Type 1, 2, and 3 and not include other newer descriptive terms of cell death and they stated "type 2 and 3 are characterized by cytoplasmic changes such as organelle vacuolation without remarkable changes of nucleus, and therefore both are included in the concept of necrosis". Such a framework requires that they only need to exclude apoptosis, which was achieved by showing this new Extended Data Figure 6.

There are a few remaining concerns raised by the new data provided .

1. The authors added new images as Extended Data Figure 5c,d to show that a "dark cell" and a pSer46-MARCKS positive DN are separate structures even if they have a close spatial relationship. On the other hand, in Extended Data Figure 5a, "the center" was considered by the authors as representing "the death process of a neuron" (Rebuttal file, Page 4, Line 7) and the organelles found there "represented the remnants of a ruined cell" (Extended Data Figure 5 legend). Given that similar "dark neurons" have been described in many cases in the literature in the absence of known mechanisms, can the authors clarify why such dark cells should be included? Is there an implication that such dark cells ["non-apoptotic dying neurons (no chromatin condensation or apoptotic body) with a deforming and shrinking but nucleus" – L227-228] are the origins of the "ruined cell" generated by "active necrosis" in "the center"? If yes, then evidence should be provided for the sequence of events that would be involved in transition of the dark cell to the necrotic pattern shown by the degenerated cell to let readers understand the significance of presenting such cells. If not, please clarify the relationship to avoid a possible misunderstanding that "dark neurons" are undergoing the "active necrosis" process described in this paper. One possible solution would be to exclude Extended Data Figure 5c,d).

2. In response to the comment regarding Fig 5a (previous Fig. 4a), the authors stated that "the DAPI signals were normalized and adjusted among multiple panels by using DAPI signals of glia cells". Was this normalization only done in the MCI samples given that only the DAPI signals in the MCI image have been changed (dimmed) compared to its previous version while DAPI signals in the Control and AD images appear unadjusted (so no changes were made compared to their previous version)? If so, the same type of normalization should be done for the Control and AD samples to establish more accurately whether or not there are differences between MCI and Control or MCI and AD in terms of DAPI intensity.

There is the concern that when the correction is applied uniformly to the three types of samples, there may not be much difference in the DAPI signal intensity and the nuclear morphology among the three types of samples, as was commented upon in the previous review.

Reviewers' comments:

Reviewer #3 (Remarks to the Author):

The authors have made significant efforts to address the remaining concerns, including a new Extended Data Figure 6 of neuronal cultures showing that “necrosis but not apoptosis could induce pSer46-MARCKS in surrounding cells” which supports their view that pSer46-MARCKS is a specific marker for necrosis when combined with the DAPI signal for identifying the “active necrosis”. They made their classification of cell death simpler by citing Schweichel and Merker, 1973; Clarke 1990 to just mention Type 1, 2, and 3 and not include other newer descriptive terms of cell death and they stated “type 2 and 3 are characterized by cytoplasmic changes such as organelle vacuolation without remarkable changes of nucleus, and therefore both are included in the concept of necrosis”. Such a framework requires that they only need to exclude apoptosis, which was achieved by showing this new Extended Data Figure 6.

There are a few remaining concerns raised by the new data provided .

1. The authors added new images as Extended Data Figure 5c,d to show that a “dark cell” and a pSer46-MARCKS positive DN are separate structures even if they have a close spatial relationship. On the other hand, in Extended Data Figure 5a, “the center” was considered by the authors as representing “the death process of a neuron” (Rebuttal file, Page 4, Line 7) and the organelles found there “represented the remnants of a ruined cell” (Extended Data Figure 5 legend).

>>> In the previous rebuttal file (Page 4, from Line 7 – 9), we described, “in Ext Data Fig 5a, the death process of a neuron at the center is *more advanced than the surrounding neurites of other neurons*”

Given that similar “dark neurons” have been described in many cases in the literature in the absence of known mechanisms, can the authors clarify why such dark cells should be included? Is there an implication that such dark cells [“non-apoptotic dying neurons (no chromatin condensation or apoptotic body) with a deforming and shrinking but nucleus” – L227-228] are the origins of the “ruined cell” generated by “active necrosis” in “the center”?

>>> We appreciate very much this interesting viewpoint from the reviewer. As the reviewer may know, “dark neuron” seemed observed in 1903 already (Turner, Brain 1903), considered as an artifact (for instance Cammermeyer, Acta Neuropathol 1961), and has been discussed over a long time on whether the artifact could reflect some changes in vivo (Ebels, Acta Neuropathol 1975). Ebels described that “dark neurons” were related to age of rats though it was blocked by rapid fixation, and concluded that it is related to something in vivo.

In addition, several ways of histological detection have been described. Gallyas reported “dark neurons” as an argyrophilic neuron in the silver stain (Gallyas et al, Acta Neuropathol 1992), though others frequently used Nissl or HE.

Following papers, for example (Colbourne et al, J Neurosci 1999), showed EM images of neurons with high electron densities, while the link among Gallyas/HE/Nissl stains and EM electron density was not exactly shown.

Some homologous morphologies have been reported in AD model (Chui et al, Nat Med 1999), Huntington’s disease model (Turmaine et al, PNAS 2000) and so on. These authors link the dark neurons to phenomena in vivo. We believe fixation had been adequate in these cases by the authorities.

(Turmaine et al, PNAS, 2000)

(Colbourne et al, J Neurosci 1999)

On the other hand, some authors again claimed relatively recently that “dark neurons” is a histological artifact. For instance, please have a look at the paper of Jortner (Neurotoxicology 2006).

<https://www.sciencedirect.com/science/article/pii/S0161813X06000490?via%3Dihub>

Therefore, collectively, the definition of “dark neurons” still remain vague and the etiology could be heterogenous.

In any case, though it is not certain whether all types of “dark neurons” are histological artifacts or not, our data (live imaging in vivo and in vitro, life reaction such as degenerative neurites full of autophagosome and MARCKS phosphorylation, etc) definitely prove that “active necrosis” is not such an artifact.

In addition, “dark neurons” are more electron-dense than our “active necrosis” (Ext Data Figure 5c, d), “dark neurons” are shrinking while “active necrosis” is an expanding rupture of the cytoplasm, “dark neurons” do not have so large cytoplasmic vacuoles like active necrosis, and “dark neurons” do not accompany biological reactions that occur during the object is alive such as degenerative neurites surrounding “active necrosis”.

We are not sure how the reviewer evaluates “dark neurons” and on which position the reviewer stands. However, the challenge to link “dark neurons” and “active necrosis” would not provide us definite meaning unless the definition and the mechanism of “dark neurons” have been fixed.

If yes, then evidence should be provided for the sequence of events that would involved in transition of the dark cell to the necrotic pattern shown by the degenerated cell to let readers understand the significance of presenting such cells.

>>> We cannot say “yes” currently as described above. And linking the two definitions of cell death is far beyond the scope of this manuscript.

If not, please clarify the relationship to avoid a possible misunderstanding that “dark neurons” are undergoing the “active necrosis” process described in this paper. One possible solution would be to exclude Extended Data Figure 5c,d).

>>> Following the advice, we delete the previous Extended Data Figure 5c,d.

However, it does not mean excluding the possibility that future studies from our group or the other groups might reveal a kind of “dark neurons” that is not an artifact is related to the “active necrosis” or “secondary necrosis”.

2. In response to the comment regarding Fig 5a (previous Fig. 4a), the authors stated that “the DAPI signals were normalized and adjusted among multiple panels by using DAPI signals of glia cells”. Was this normalization only done in the MCI samples given that only the DAPI signals in the MCI image have been changed (dimmed) compared to its previous version while DAPI signals in the Control and AD images appear unadjusted (so no changes were made compared to their previous version)? If so, the same type of normalization should be done for the Control and AD samples to establish more accurately whether or not there are differences between MCI and Control or MCI and AD in term of DAPI intensity.

>>> Though the images might look similar, we have done normalization with glial DAPI in the Control and AD images in the previous version already.

There is the concern that when the correction is applied uniformly to the three type of samples, there may not be much difference in the DAPI signal intensity and the nuclear morphology among the three type of samples, as was commented upon in the previous review.

>>> We now added another panel to Figure 5b that compares DAPI signals among Control, MCI and AD. The result supported the decrease of DAPI signals in neurons showing colocalization of YAP and Abeta in the cytoplasm.

REVIEWERS' COMMENTS:

Reviewer #3 (Remarks to the Author):

The authors' responses are satisfactory and there are no further concerns before publication.